# A Near-optimal, Scalable and Parallelizable Framework for Stochastic Bandits Robust to Adversarial Corruptions and Beyond

**Zicheng Hu**    **Cheng Chen**[*]
MoE Engineering Research Center of Hardware/Software Co-design Technology and Application
East China Normal University
51275902019@stu.ecnu.edu.cn    chchen@sei.ecnu.edu.cn

## Abstract

We investigate various stochastic bandit problems in the presence of adversarial corruptions. A seminal work for this problem is the BARBAR [1] algorithm, which achieves both robustness and efficiency. However, it suffers from a regret of $O(KC)$, which does not match the lower bound of $\Omega(C)$, where $K$ denotes the number of arms and $C$ denotes the corruption level. In this paper, we first improve the BARBAR algorithm by proposing a novel framework called BARBAT, which eliminates the factor of $K$ to achieve an optimal regret bound up to a logarithmic factor. We also extend BARBAT to various settings, including multi-agent bandits, graph bandits, combinatorial semi-bandits and batched bandits. Compared with the Follow-the-Regularized-Leader framework, our methods are more amenable to parallelization, making them suitable for multi-agent and batched bandit settings, and they incur lower computational costs, particularly in semi-bandit problems. Numerical experiments verify the efficiency of the proposed methods.

## 1 Introduction

The multi-armed bandit (MAB) problem is one of the most fundamental online learning tasks, and it has a long and rich history [2, 3]. Recently, as security concerns have grown, many studies focus on the so-called adversarially corrupted regime where an adversary is allowed to manipulate reward observations in a stochastic environment. Existing approaches can be broadly categorized into two classes: the Follow-the-Regularized-Leader (FTRL) family of algorithms [4, 5, 6, 7] and elimination-based algorithms [8, 1, 9, 10]. FTRL-based methods perform optimally in both stochastic and adversarial settings and also achieve optimal regret in the adversarially corrupted setting. However, these methods need to solve an optimization problem in each round, which may incur high computational costs in many scenarios such as semi-bandits. Also, the FTRL framework is challenging to extend to parallel settings such as batched bandits and multi-agent bandits. On the other hand, elimination-based methods such as BARBAR [1] are computationally efficient and easy to parallelize. However, the BARBAR algorithm suffers from a suboptimal regret of $O(KC)$, and it remains an open question how to achieve optimal regret by elimination-based methods.

In this paper, we improve upon the BARBAR [1] method and propose a novel framework called BARBAT (**B**ad **A**rms get **R**ecourse, **B**est **A**rm gets **T**rust) for stochastic bandits with adversarial corruptions. Unlike BARBAR, which employs dynamic epoch lengths, BARBAT adopts static epoch lengths by increasing the probability of selecting the estimated best arm. In addition, BARBAT utilizes epoch-varying failure probabilities $\delta_m$ instead of a global failure probability $\delta$ used in BARBAR, allowing us to achieve near-optimal regret bounds. We further demonstrate that our

---

[*]The corresponding author

39th Conference on Neural Information Processing Systems (NeurIPS 2025).

BARBAT framework is scalable and parallelizable by extending it to various scenarios, including cooperative multi-agent multi-armed bandits (CMA2B) [11, 9, 12], graph bandits [13, 14], semi-bandits [15, 16, 17] and batched bandits [18, 19, 20]. The regret bounds of our methods along with comparisons to recent works are summarized in Table 1.

We summarize the contributions of this work as follows:

- **BARBAT framework**    We propose the BARBAT framework for stochastic bandits with adversarial corruptions that achieves a near-optimal regret bound. This result resolves the open problem raised in [1]. Moreover, BARBAT does not require prior knowledge of the time horizon $T$.

- **Extensions of BARBAT**    We extend our BARBAT framework to various corrupted bandit settings including multi-agent bandits, batched bandits, strongly observable graph bandits and $d$-set semi-bandits. We show that all resulting algorithms achieve near-optimal regret. To the best of our knowledge, our work is the first to study a robust algorithm for batched bandits. We also provide a lower bound for batched bandits with adversarial corruptions.

- **Comparison with FTRL**    Compared to FTRL-based methods, our framework has advantages in three ways. First, our methods are more efficient. The FTRL framework requires solving a constrained convex optimization problem in each round. Such problems generally do not have a closed-form solution (except for FTRL with Shannon-entropy regularization) and can therefore be computationally costly, particularly in semi-bandit scenarios. Second, our BARBAT framework is parallelizable, making it suitable for batched and multi-agent settings, whereas extending FTRL to these scenarios is quite challenging. Lastly, as shown in the last column of Table 1, our methods do not require the assumption of a unique optimal action, which is required by FTRL-based methods in environments other than MAB.

**Related work**    In recent years, the study of stochastic bandits with adversarial corruptions has garnered significant attention. Existing approaches can be broadly classified into two main categories: elimination-based algorithms [8, 1, 9, 10] and FTRL-based algorithms [4, 5, 6, 7]. Elimination-based methods, with the Active Arm Elimination (AAE) algorithm [21] being a prominent example, progressively discard suboptimal arms based on empirical reward estimates. By removing poorly performing arms, these algorithms focus their sampling efforts on a shrinking set of promising candidates. [8] introduced a multi-layer extension of the AAE algorithm, achieving a regret bound of $\widetilde{O}\left(KC\sum_{\Delta_k>0} 1/\Delta_k\right)$. Building on this, [1] proposed the BARBAR algorithm and reduced the regret to $\widetilde{O}\left(KC + \sum_{\Delta_k>0} 1/\Delta_k\right)$. On the other hand, FTRL-based methods [22, 23, 24, 25, 4, 5, 6, 7] can achieve optimal regret bounds in the stochastic, adversarial and adversarially corrupted settings. However, these methods are more computationally expensive than elimination-based methods since they need to solve an optimization problem in each round.

Stochastic bandits with adversarial corruptions can be extended to graph bandits, semi-bandits, and batched bandits, each of which benefits from specialized techniques to handle challenges such as exponential action spaces, partial observability, or limited adaptivity [26, 13, 14, 15, 16, 17, 18, 19, 20]. Finally, multi-agent extensions [9, 11, 27, 12] enable agents to pool information and expedite the learning process but often encounter larger regret bounds under adversarial corruptions or rely on group-level performance guarantees rather than individual regrets.

## 2    Preliminaries

We consider stochastic multi-armed bandits with adversarial corruptions. In this setting, the agent interacts with the environment over $T$ rounds by selecting an arm from a set of $K$ arms, denoted by $[K]$. In each round $t$, the environment generates an i.i.d. random reward vector $\{r_{t,k}\}_{k\in[K]}$ from an unknown fixed distribution. An adversary, having access to the reward vector, subsequently attacks these rewards to produce the corrupted reward vector $\{\widetilde{r}_{t,k}\}_{k\in[K]}$. The agent then selects an arm $I_t$ according to its strategy and observes the corresponding corrupted reward $\widetilde{r}_{t,I_t}$. Let $\mu_k$ denote the mean reward of arm $k \in [K]$, and let $k^* \in \arg\max_{k\in[K]} \mu_k$ be an optimal arm. The corruption level is defined as $C = \sum_{t=1}^{T} \max_{k\in[K]} |\widetilde{r}_{t,k} - r_{t,k}|$. The suboptimality gap for arm $k$ is defined as $\Delta_k = \mu_k - \mu_{k^*}$, and we denote $\Delta = \min_{\Delta_k>0} \Delta_k$ as the smallest positive suboptimality gap.

| Setting | Algorithm | Regret | Prop. 1 | Prop. 2 |
|---|---|---|---|---|
| Multi-arm bandits | BARBAR [1] | $KC + \sum_{\Delta_k>0} \frac{\log^2(T)}{\Delta_k}$ | ✓ | ✓ |
| | $\frac{1}{2}$-Tsallis-FTRL [4] | $C + \sum_{\Delta_k>0} \frac{\log(T)}{\Delta_k}$ | ✗ | ✗ |
| | $\frac{1}{2}$-Tsallis-FTRL [6] | $C + \sum_{\Delta_k>0} \frac{\log(T)}{\Delta_k}$ | ✗ | ✓ |
| | Shannon-FTRL [6, 13] | $C + \sum_{\Delta_k>0} \frac{\log^2(T)}{\Delta_k}$ | ✓ | ✓ |
| | Algorithm 1 | $C + \sum_{\Delta_k>0} \frac{\log^2(T)}{\Delta_k}$ | ✓ | ✓ |
| | Lower bound [1] | $C + \sum_{\Delta_k>0} \frac{\log(T)}{\Delta_k}$ | – | – |
| Multi-agent multi-arm bandits | CBARAC [9] | $C + \frac{K \log^2(T)}{\Delta}$ | ✓ | ✓ |
| | DRAA [12] | $\frac{C}{V} + \frac{K \log^2(T)}{V\Delta}$ | ✓ | ✓ |
| | Algorithm 2 | $\frac{C}{V} + \sum_{\Delta_k>0} \frac{\log^2(T)}{V\Delta_k}$ | ✓ | ✓ |
| | Lower bound‡ | $\frac{C}{V} + \sum_{\Delta_k>0} \frac{\log(T)}{V\Delta_k}$ | – | – |
| Batched bandits | Algorithm 3 | $CT^{\frac{1}{L+3}} + T^{\frac{4}{L+3}}\left(\sum_{\Delta_k>0} \frac{L\log(T)}{\Delta_k} + \frac{K\log(T)}{L\Delta}\right)$ | ✓ | ✓ |
| | Lower bound (Ours) | $T^{1/L}(K + C^{1-1/L})$ | – | – |
| Strongly observable graph bandits† | Elise [10] | $\alpha C + \sum_{k\in\mathcal{D}^*} \frac{\log^2(T)}{\Delta_k}$ | ✓ | ✓ |
| | Shannon-FTRL [13] | $C + \frac{\alpha \log^3(T)}{\Delta}$ | ✓ | ✗ |
| | $(1-\frac{1}{\log(K)})$-Tsallis-FTRL[14] | $C + \frac{\min\{\widetilde{\alpha}, \alpha\log(K)\}\log(T)}{\Delta}$ | ✗ | ✗ |
| | Algorithm 4 | $C + \sum_{k\in\mathcal{I}^*} \frac{\log^2(T)}{\Delta_k}$ | ✓ | ✓ |
| | Lower bound‡ | $C + \sum_{k\in\mathcal{I}^*} \frac{\log(T)}{\Delta_k}$ | – | – |
| $d$-set semi-bandits | HYBRID [15] | $dC + \sum_{\Delta_k>0} \frac{\log(T)}{\Delta_k}$ | ✗ | ✗ |
| | LBINF [16] | $dC + \sum_{\Delta_k>0} \frac{\log(T)}{\Delta_k}$ | ✗ | ✗ |
| | LBINF_LS, LBINF_GD [17] | $dC + \sum_{\Delta_k>0} \frac{\log(T)}{\Delta_k}$ | ✗ | ✗ |
| | Algorithm 5 | $dC + \sum_{\Delta_k>0} \frac{\log^2(T)}{\Delta_k}$ | ✓ | ✓ |
| | Lower bound‡ | $dC + \sum_{\Delta_k>0} \frac{\log(T)}{\Delta_k}$ | – | – |

Table 1: "Prop. 1" denotes that the algorithm can efficiently compute the sampling probabilities of the arms, while "Prop. 2" denotes that the assumption of a unique optimal arm is not required. For graph bandits, $\alpha$ denotes the *independence number* of graph $G$ and $\mathcal{I}^*$ denotes the set of at most $O\big(\alpha \ln\big(\frac{K}{\alpha} + 1\big)\big)$ arms with the smallest gaps. For batched bandits, $L$ denotes the number of batches. "†": Elise is restricted to undirected graphs, while the remaining studies address directed graphs. "‡": These lower bounds can be directly achieved by combining the corruption lower bound from [1] with the lower bound for the stochastic setting.

Our goal is to minimize the pseudo-regret:

$$R(T) = \max_{k\in[K]} \mathbb{E}\left[\sum_{t=1}^{T} r_{t,k}\right] - \mathbb{E}\left[\sum_{t=1}^{T} r_{t,I_t}\right] = \sum_{t=1}^{T} \mu_{k^*} - \sum_{t=1}^{T} \mu_{I_t} = \sum_{t=1}^{T} \Delta_{I_t},$$

which is a standard definition in stochastic bandits. Notice that FTRL-based methods [26, 13, 14, 15, 16, 17, 18, 19, 20] typically adopt a different form of pseudo-regret, denoted as $\widetilde{R}(T) = \max_k \mathbb{E}\big[\sum_{t=1}^{T} (\tilde{r}_{t,I_t} - \tilde{r}_{t,k})\big]$, which is more appropriate for adversarial settings. However, we argue

that comparing against the true means, rather than the corrupted rewards, is more natural. Theorem 3 in [9] presents the conversion between the two definitions of pseudo-regret in the adversarially corrupted setting as $R(T) = \Theta(\widetilde{R}(T) + O(C))$.

**CMA2B**  In cooperative multi-agent multi-armed bandits, $V$ agents collaborate to play the bandit game. We denote the set of agents by $[V]$. All agents are allowed to pull any arm from $[K]$. When multiple agents pull the same arm, each receives an independent reward drawn from that arm's distribution. The adversary is allowed to attack all agents with a total corruption budget $C$. We only consider the centralized setting where each agent can broadcast messages to all other agents without any delays. For simplicity, we define the total communication cost for each agent $v$ as:

$$\text{Cost}_v(T) \triangleq \sum_{v \in [V]} \sum_{t=1}^{T} \mathbb{I}\{\text{agent } v \text{ broadcasts a message to other agents in round } t\}.$$

We aim to minimize the individual pseudo-regret for each agent $v$ over a horizon of $T$ rounds:

$$R_v(T) = \max_{k \in [K]} \mathbb{E}\left[\sum_{t=1}^{T} r_{v,t}(k)\right] - \mathbb{E}\left[\sum_{t=1}^{T} r_{v,t}(I_{v,t})\right] = \sum_{t=1}^{T}(\mu_{k^*} - \mu_{I_{v,t}}) = \sum_{t=1}^{T} \Delta_{I_{v,t}}.$$

**Batched bandits**  In this case, we suppose the agent can only observe the corrupted rewards from a batch after it has concluded. The time horizon $T$ is divided into $L$ batches, represented by a grid $\mathcal{T} = \{t_1, t_2, \ldots, t_L\}$, where $1 \leq t_1 < t_2 < \cdots < t_L = T$. As in most previous works [19], the number of batches $L$ of interest is at most $\log(T)$. Additionally, the grid can be one of two types:

- **Static Grid:** The grid is fixed and predetermined before any arm sampling occurs.
- **Adaptive Grid:** The value of $t_j$ is determined dynamically after observing the rewards up to time $t_{j-1}$, which may incorporate external randomness.

Previous works [18, 19, 28, 20, 29] mainly focus on stochastic batched bandits. To the best of our knowledge, no existing work studies stochastic batched bandits under adversarial corruptions.

**Strongly observable graph bandits**  In this setting, when the agent pulls arm $k$, it may observe the rewards of other arms. This reward-feedback structure is represented by a directed graph $G = ([K], E)$, where pulling arm $k_i$ reveals the rewards of each arm $k_j$ such that $(k_i, k_j) \in E$. Each arm either has a self-loop or receives incoming edges from all other arms [30]. Let $\alpha$ denote the independence number of $G$, i.e., the maximum size of an independent set. A vertex set $D$ is called an out-domination set of $G$ if every vertex in $G$ has an incoming edge from some vertex in $D$.

$d$**-set semi-bandits**  The $d$-set semi-bandits problem is a special case of semi-bandits. Given $d \in [K-1]$, in each round $t$ the agent selects a combinatorial action $\mathbf{x}_t$ (containing $d$ distinct arms) from the set $\mathcal{X} = \left\{\mathbf{x} \in \{0,1\}^K \mid \sum_{k=1}^{K} \mathbf{x}(k) = d\right\}$, where $\mathbf{x}(k) = 1$ indicates that arm $k$ is selected. Then the agent can observe the corrupted rewards for each arm in the action (i.e., for each arm $k$ such that $\mathbf{x}_t(k) = 1$). Let $\mu_k$ denote the mean reward of arm $k \in [K]$. Without loss of generality, we assume $\mu_1 \geq \mu_2 \geq \cdots \geq \mu_K$ and denote $\Delta_k = \mu_d - \mu_k$ for all $k > d$. We aim to minimize the pseudo-regret over a horizon of $T$ rounds:

$$R(T) = \max_{\mathbf{x} \in \mathcal{X}} \mathbb{E}\left[\sum_{t=1}^{T} \sum_{k:\mathbf{x}[k]=1} r_{t,k}\right] - \mathbb{E}\left[\sum_{t=1}^{T} \sum_{k:\mathbf{x}_t[k]=1} r_{t,k}\right] = \sum_{t=1}^{T}\left(\sum_{k=1}^{d} \mu_k - \sum_{k:\mathbf{x}_t(k)=1} \mu_k\right).$$

## 3  Stochastic Multi-Armed Bandits Robust to Adversarial Corruptions

**Main idea**  Notice that in the BARBAR [1] algorithm, the arm $k$ is approximately drawn $n_k^m = O(1/(\Delta_k^{m-1})^2)$ times in expectation during epoch $m$, where $\Delta_k^{m-1}$ is the estimated gap computed in the previous epoch. Thus, each epoch has a length of approximately $N_m \approx O(\sum_k 1/(\Delta_k^{m-1})^2)$. Consequently, the adversary can significantly extend the length of an epoch by utilizing all of the corruption budget in the previous epoch, thereby implicitly increasing the number of pulls for

---

**Algorithm 1** BARBAT: Bad Arms get Recourse, Best Arm gets Trust

---

1: **Initialization:** Set the initial round $T_0 \leftarrow 0$, $\Delta_k^0 \leftarrow 1$, and $r_k^0 \leftarrow 0$ for all $k \in [K]$.
2: **for** epochs $m = 1, 2, \cdots$ **do**
3:      Set $\zeta_m \leftarrow (m+4)2^{2(m+4)} \ln(K)$ and $\delta_m \leftarrow 1/(K\zeta_m)$.
4:      Set $\lambda_m \leftarrow 2^8 \ln(4K/\delta_m)$ and $\beta_m \leftarrow \delta_m/K$.
5:      Set $n_k^m \leftarrow \lambda_m(\Delta_k^{m-1})^{-2}$ for all $k \in [K]$.
6:      Set $N_m \leftarrow \lceil K\lambda_m 2^{2(m-1)} \rceil$ and $T_m \leftarrow T_{m-1} + N_m$.
7:      Set $k_m \leftarrow \arg\max_{k \in [K]} r_k^{m-1}$ and compute $\widetilde{n}_k^m$ according to (1).
8:      **for** $t = T_{m-1} + 1$ to $T_m$ **do**
9:          Choose arm $I_t \sim p_m$ where $p_m(k) = \widetilde{n}_k^m/N_m$.
10:          Observe the corrupted reward $\widetilde{r}_{t,I_t}$ and update the total reward $S_{I_t}^m \leftarrow S_{I_t}^m + \widetilde{r}_{t,I_t}$.
11:      **end for**
12:      Set $r_k^m \leftarrow \min\{S_k^m/\widetilde{n}_k^m, 1\}$ and $r_*^m \leftarrow \max_{k \in [K]} \left\{ r_k^m - \sqrt{\frac{4\ln(4/\beta_m)}{\widetilde{n}_k^m}} \right\}$.
13:      Set $\Delta_k^m \leftarrow \max\{2^{-m}, r_*^m - r_k^m\}$.
14: **end for**

---

suboptimal arms. To address this challenge, our BARBAT algorithm chooses a data-independent epoch length $N_m \approx O(K2^{2(m-1)})$, which cannot be affected by the adversary. We still allocate $n_k^m = O(1/(\Delta_k^{m-1})^2)$ pulls to each arm $k$, but assign all remaining pulls to the estimated optimal arm $k_m$, resulting in pulling arm $k_m$ for about $N_m - \sum_{k \neq k_m} n_k^m$ rounds. On the other hand, pulling the estimated optimal arm $k_m$ too many times may lead to additional regret, as arm $k_m$ could be suboptimal. To mitigate this additional regret, we adopt epoch-varying failure probabilities $\delta_m \approx O(1/(mK2^{2m}))$ for epoch $m$ rather than using a global failure probability $\delta$. The choice of $\delta_m$ also eliminates the need to know the time horizon $T$.

**Algorithm**    We introduce our BARBAT framework for stochastic bandits with adversarial corruptions in Algorithm 1. In each epoch $m$, BARBAT sets the failure probability as $\delta_m = 1/(K\zeta_m)$, where $\zeta_m$ is chosen to make $\delta_m \leq 1/N_m$. The algorithm sets the epoch length to be $N_m = \lceil K\lambda_m 2^{2(m-1)} \rceil \geq \sum_{k \in [K]} n_k^m$. We denote $k_m$ as the arm with the maximum empirical reward in the previous epoch; in case of ties, the arm with the smallest index is chosen. Then the number of pulls for each arm is set to

$$\widetilde{n}_k^m = \begin{cases} n_k^m, & k \neq k_m, \\ N_m - \sum_{k \neq k_m} n_k^m, & k = k_m. \end{cases} \tag{1}$$

During epoch $m$, the agent pulls arm $I_t$ with probability $p_m(I_t) = \widetilde{n}_{I_t}^m/N_m$, observes the corrupted reward $\widetilde{r}_{t,I_t}$, and accumulates the total reward $S_{I_t}^m$. At the end of each epoch $m$, BARBAT updates the estimated reward $r_k^m = \min\{S_k^m/\widetilde{n}_k^m, 1\}$ and computes the estimated suboptimality gap $\Delta_k^m$.

**An example showing how BARBAT avoids paying an $O(KC)$ term**    Gupta et al. [1] provide an example where BARBAR suffers regret $\Omega(KC)$, in which all corruptions happen in an epoch $c$ with corruption level $C = N_c = \lambda(2^{2(c-1)} + O(K))$. Then the algorithm will lose all information from the previous epochs so that the length of the next epoch becomes $N_{c+1} = K\lambda 2^{2c} = O(KC)$ and each arm will be pulled uniformly. In this scenario, BARBAR will pull each arm approximately $O(C)$ times and generate $O((K-1)C)$ regret. On the other hand, our BARBAT algorithm fixes the length of the $m$-th epoch to be $N_m = K\lambda_m 2^{2(m-1)}$. If the adversary intends to corrupt an entire epoch $c$, the required corruption level is $C = N_c = K\lambda 2^{2(c-1)}$, which is $K$ times larger than needed to attack BARBAR. Thus our BARBAT algorithm will only suffer $O(C)$ regret from adversarial corruptions.

**Regret Bound**    BARBAT achieves the following regret bound, with proofs deferred to Appendix B.1.

**Theorem 1.** *The expected regret of BARBAT satisfies*

$$R(T) = O\left(C + \sum_{\Delta_k > 0} \frac{\log(T)\log(KT)}{\Delta_k} + \frac{K\log(1/\Delta)\log(K/\Delta)}{\Delta}\right).$$

---

**Algorithm 2** MA-BARBAT
---
1: **Initialization:** Set the initial round $T_0 \leftarrow 0$, $\Delta_k^0 \leftarrow 1$, and $r_k^0 \leftarrow 0$ for all $k \in [K]$.
2: **for** epoch $m = 1, 2, \ldots$ **do**
3:     Set $\zeta_m \leftarrow (m+4)2^{2(m+4)} \ln(VK)$ and $\delta_m \leftarrow 1/(VK\zeta_m)$.
4:     Set $\lambda_m \leftarrow 2^8 \ln\left(4K/\delta_m\right)/V$ and $\beta_m \leftarrow \delta_m/(VK)$.
5:     Set $n_k^m \leftarrow \lambda_m(\Delta_k^{m-1})^{-2}$ for all $k \in [K]$.
6:     Set $N_m \leftarrow \lceil K\lambda_m 2^{2(m-1)} \rceil$ and $T_m \leftarrow T_{m-1} + N_m$.
7:     Set $k_m \leftarrow \arg\max_{k \in [K]} r_k^{m-1}$ and compute $\widetilde{n}_k^m$ according to (1).
8:     **for** $t = T_{m-1} + 1$ **to** $T_m$ **do**
9:         Each agent chooses arm $I_t \sim p_m$, where $p_m(k) = \widetilde{n}_k^m/N_m$.
10:         Each agent observes the corrupted reward $\widetilde{r}_{t,I_t}$ and compute $S_{I_t}^m \leftarrow S_{I_t}^m + \widetilde{r}_{t,I_t}$.
11:     **end for**
12:     Each agent $v$ broadcasts the messages $\{S_{v,k}^m\}_{k \in [K]}$ to other agents.
13:     Set $r_k^m \leftarrow \min\{\sum_{v \in [V]} S_{v,k}^m/(V\widetilde{n}_k^m), 1\}$ and $r_*^m \leftarrow \max_{k \in [K]} \left\{ r_k^m - \sqrt{\frac{4\ln(4/\beta_m)}{V\widetilde{n}_k^m}} \right\}$.
14:     Set $\Delta_k^m \leftarrow \max\{2^{-m}, r_*^m - r_k^m\}$.
15: **end for**
---

**Remark 1.** *The regret bound of BARBAT contains an extra term $K\log^2(1/\Delta)/\Delta$. Notice that this term is independent of $T$, making it generally much smaller than the major term $\sum_{\Delta_k>0} \frac{\log^2(T)}{\Delta_k}$.*

**Remark 2.** *Notice that a concurrent work [12] also employs static epoch lengths and removes the factor $K$ from the regret bound. However, they adopt a global $\delta$ across all epochs, resulting in a regret of $O(C + K\log^2(T)/\Delta)$, which is worse than our regret of $O\left(C + \sum_{\Delta_k>0} \frac{\log^2(T)}{\Delta_k}\right)$.*

## 4 Extensions

In this section, we extend BARBAT to various corrupted bandit settings and introduce the necessary modifications to accommodate the specific configurations of these environments.

### 4.1 Cooperative Multi-Agent Multi-Armed Bandits

We extend BARBAT to the multi-agent setting and propose the MA-BARBAT (**M**ulti-**A**gent **BAR-BAT**) algorithm in Algorithm 2. In each epoch $m$, all agents pull arms according to the same probability distribution and broadcast their total rewards $\{S_{v,k}^m\}_{k=1}^K$ at the end of the epoch. Each agent then updates the estimated suboptimality gaps $\Delta_k^m$ using the received messages and its own observations. We present the regret bound and communication cost of our MA-BARBAT algorithm in the following theorem, with the proof given in Appendix B.2.

**Theorem 2.** *The MA-BARBAT algorithm requires only a communication cost of $V\log(VT)$ over $T$ rounds. The individual regret of each agent $v$ satisfies:*

$$R_v(T) = O\left( \frac{C}{V} + \sum_{\Delta_k>0} \frac{\log(VT)\log(KVT)}{V\Delta_k} + \frac{K\log(1/\Delta)\log(KV/\Delta)}{V\Delta} \right).$$

**Remark 3.** *The regret bound demonstrates that collaboration reduces each agent's individual regret by a factor of $V$. As summarized in Table 1, our bound is tighter than those of previous works on CMA2B with adversarial corruptions.*

**Remark 4.** *Notice that some recent works [11] on CMA2B achieve a communication cost of only $o(\log(T))$. However, these methods rely on the assumption of a unique best arm. In the presence of multiple best arms, it remains unclear whether a communication cost of $o(\log(T))$ is sufficient.*

### 4.2 Batched Bandits

We extend BARBAT to the batched bandit setting and propose the BB-BARBAT (**B**atched **B**andits-**BARBAT**) algorithm in Algorithm 3. The BB-BARBAT algorithm sets the epoch lengths $N_m$ to

---

**Algorithm 3** BB-BARBAT

---

1: **Initialization:** Set the initial round $T_0 \leftarrow 0$, $\Delta_k^0 \leftarrow 1$, and $r_k^0 \leftarrow 0$ for all $k \in [K]$.
2: **for** epochs $m = 1, 2, \cdots$ **do**
3:      Set $a \leftarrow T^{\frac{1}{2(L+1)}}$, $\zeta_m \leftarrow (m+4)a^{2(m+4)} \ln(aK)$ and $\delta_m \leftarrow 1/(K\zeta_m)$.
4:      Set $\lambda_m \leftarrow a^8 \ln(4K/\delta_m)$ and $\beta_m \leftarrow \delta_m/K$.
5:      Set $n_k^m \leftarrow \lambda_m (\Delta_k^{m-1})^{-2}$ for all $k \in [K]$.
6:      Set $N_m \leftarrow \lceil K\lambda_m a^{2(m-1)} \rceil$ and $T_m \leftarrow T_{m-1} + N_m$.
7:      Set $k_m \leftarrow \arg\max_{k \in [K]} r_k^{m-1}$ and compute $\widetilde{n}_k^m$ according to (1).
8:      **for** $t = T_{m-1} + 1$ to $T_m$ **do**
9:          Choose arm $I_t \sim p_m$ where $p_m(k) = \widetilde{n}_k^m / N_m$.
10:      **end for**
11:      Observe the all corrupted rewards $\widetilde{r}_{I_t}$ in this batch and compute $S_{I_t}^m \leftarrow S_{I_t}^m + \widetilde{r}_{I_t}$.
12:      Set $r_k^m \leftarrow \min\{S_k^m / \widetilde{n}_k^m, 1\}$ and $r_*^m \leftarrow \max_{k \in [K]} \left\{ r_k^m - \sqrt{\frac{4\ln(4/\beta_m)}{\widetilde{n}_k^m}} \right\}$.
13:      Set $\Delta_k^m \leftarrow \max\{a^{-m}, r_*^m - r_k^m\}$.
14: **end for**

---

be approximately $O(T^{\frac{m}{L+1}})$ ensuring that the number of epochs matches the number of batches. In this way, each epoch serves as a batch, making the BB-BARBAT algorithm almost identical to the BARBAT algorithm. We present the regret upper bound of BB-BARBAT and the lower bound for corrupted batched bandits in Theorems 3 and 4, respectively. The proofs are deferred to Appendix B.3.

**Theorem 3.** *For corrupted batched bandits with a static grid, the regret of BB-BARBAT satisfies*

$$R(T) = O\left( CT^{\frac{1}{L+3}} + T^{\frac{4}{L+3}} \left( \sum_{\Delta_k > 0} \frac{L\log(KT)}{\Delta_k} + \frac{K\log(T)\log(1/\Delta)\log(K/\Delta)}{L\Delta} \right) \right).$$

**Theorem 4.** *For any algorithm, there exists an instance of batched bandits with a static grid such that*

$$R(T) \geq \Omega\left( T^{\frac{1}{L}} \left( K + C^{1-\frac{1}{L}} \right) \right).$$

**Remark 5.** *There is still a gap between our upper bound and the lower bound. We believe that our lower bound is nearly tight. Achieving optimal regret upper bounds for corrupted batched bandits remains an interesting open question.*

## 4.3 Strongly Observable Graph Bandits

We extend BARBAT to strongly observable graph bandits and propose the SOG-BARBAT (**S**trongly-**O**bservable **G**raph bandits-**BARBAT**) algorithm, presented in Algorithm 4. Our method adopts $Z^m$ and $H^m$ to denote the expected pulls and expected observations in epoch $m$, respectively. $G^m$ denotes the graph used in the $m$-th epoch. SOG-BARBAT adopts the proposed Algorithm 6 (in Appendix A) to compute the out-domination set $D^m$ of $G^m$, whose size is at most $\alpha(1 + 2\ln(K/\alpha))$. Moreover, if $G^m$ is acyclic, we can guarantee that $|D^m| \leq \alpha$. Then SOG-BARBAT computes $\overline{H}^m$, which is the minimal number of additional pulls per out-dominating arm that ensures some vertex meets its observation requirement. After that, SOG-BARBAT updates $Z^m$ and $H^m$, and removes all nodes that have now been sufficiently observed from $G^m$. Repeating these steps ensures that by the end of epoch $m$, all arms have been observed sufficiently. The regret bound for SOG-BARBAT is presented as follows, with the proof given in Appendix B.4.

**Theorem 5.** *For any strongly observable directed graph $G = ([K], E)$ with independence number $\alpha$, the expected regret of SOG-BARBAT satisfies*

$$\mathbb{E}[R(T)] = O\left( C + \sum_{k \in \mathcal{I}^*} \frac{\log(T)\log(KT)}{\Delta_k} + \frac{K\log(1/\Delta)\log(K/\Delta)}{\Delta} \right),$$

*where $\mathcal{I}^*$ is the set of at most $\lceil \alpha(1 + 2\ln(K/\alpha)) \rceil$ arms with the smallest gaps. Especially, for directed acyclic graphs and undirected graphs, $\mathcal{I}^*$ is the set of at most $\alpha$ arms with the smallest gaps.*

**Remark 6.** *The regret upper bound of SOG-BARBAT depends on $|\mathcal{I}^*| = O(\alpha \ln(K/\alpha + 1))$, which matches the lower bound established by [31]. When $G$ is an acyclic graph or an undirected graph, we have $|\mathcal{I}^*| \leq \alpha$, indicating that SOG-BARBAT performs better under this condition.*

---

**Algorithm 4** SOG-BARBAT: Strongly Observable Graph-BARBAT

---

1: **Input:** A Strongly Observable Directed Graph $G$.
2: **Initialization:** Set the initial round $T_0 \leftarrow 0$, $\Delta_k^0 \leftarrow 1$, and $r_k^0 \leftarrow 0$ for all $k \in [K]$.
3: **for** epochs $m = 1, 2, ...$ **do**
4:     Set $\zeta_m \leftarrow (m+4)2^{2(m+4)} \ln(K)$ and $\delta_m \leftarrow 1/(K\zeta_m)$.
5:     Set $\lambda_m \leftarrow 2^8 \ln(4K/\delta_m)$ and $\beta_m \leftarrow \delta_m/K$.
6:     Set $n_k^m \leftarrow \lambda_m(\Delta_k^{m-1})^{-2}$ for all $k \in [K]$.
7:     Set $N_m \leftarrow \lceil K\lambda_m 2^{2(m-1)} \rceil$ and $T_m \leftarrow T_{m-1} + N_m$.
8:     Set $k_m \leftarrow \arg\max_{k \in [K]} r_k^{m-1}$.
9:     Set $Z_k^m \leftarrow 0$ and $H_k^m \leftarrow 0$ for all arms $k \in [K]$, $G^m \leftarrow G$.
10:     **while** $H_k^m \geq n_k^m$ holds for all arms $k \in [K]$ **do**
11:         Compute an out-domination set $D^m$ of $G^m$ by Algorithm 6.
12:         Compute $\overline{H}^m \leftarrow \min_{k \in [K]}(n_k^m - H_k^m)$.
13:         **for** each arm $k_i \in D^m$ **do**
14:             Update $Z_{k_i}^m \leftarrow Z_{k_i}^m + \overline{H}_s^m$ and $H_{k_i}^m \leftarrow H_{k_i}^m + \overline{H}^m$
15:             Update $H_{k_j}^m \leftarrow H_{k_j}^m + \overline{H}^m$ for all arms $(k_i, k_j) \in E$ with $k_j \neq k_i$
16:         **end for**
17:         Remove all arms which satisfy $H_k^m \geq n_k^m$ from $G^m$.
18:     **end while**
19:     Set $\widetilde{n}_k^m = \begin{cases} Z_k^m & k \neq k_m \\ N_m - \sum_{k \neq k_m} Z_k^m & k = k_m \end{cases}$, and $\hat{n}_{k_j}^m \leftarrow \sum_{(k_i,k_j) \in E} \widetilde{n}_{k_i}^m$.
20:     **for** $t = T_{m-1} + 1$ to $T_m$ **do**
21:         Choose arm $I_t \sim p_m$ where $p_m(k) = \widetilde{n}_k^m/N_m$.
22:         Observe the corrupted reward $\widetilde{r}_{t,I_t}$ and update the total reward $S_{I_t}^m \leftarrow S_{I_t}^m + \widetilde{r}_{t,I_t}$.
23:     **end for**
24:     Set $r_k^m \leftarrow \min\{S_k^m/\hat{n}_k^m, 1\}$ and $r_*^m \leftarrow \max_{k \in [K]} \left\{ r_k^m - \sqrt{\frac{4\ln(4/\beta_m)}{\hat{n}_k^m}} \right\}$
25:     Set $\Delta_k^m \leftarrow \max\{2^{-m}, r_*^m - r_k^m\}$.
26: **end for**

---

**Remark 7.** *Compared to FTRL-based methods [13, 14], whose regret bounds depend on the smallest suboptimality gap $1/\Delta$, our bound reveals the relationship between the regret and the suboptimality gap of all arms. In addition, these works require prior knowledge of the independence number $\alpha$, which is NP-hard to compute, whereas SOG-BARBAT does not need such prior knowledge.*

**Remark 8.** *Compared with [10], which adapts BARBAR to graph bandits with undirected graphs, our SOG-BARBAT method improves the corruption-dependent term from $O(\alpha C)$ to $O(C)$. Also, our method can be applied to bandits with directed graphs.*

## 4.4 $d$-set Semi-bandits

We extend BARBAT to the $d$-set semi-bandit setting and propose the DS-BARBAT (**d-S**et **BARBAT**) algorithm in Algorithm 5. In this setting, the optimal action is a subset of $d$ distinct arms. Thus, DS-BARBAT estimates the best action $\mathcal{K}_m$ as set of $d$ distinct arms with the highest empirical rewards in the previous epoch. When estimating the suboptimality gap, DS-BARBAT considers the arm with the $d$-th largest empirical reward, since the set of the first $d$ arms is the optimal action. The regret bound for DS-BARBAT is presented as follows, with the proof provided in Appendix B.5:

**Theorem 6.** *The regret of DS-BARBAT satisfies*

$$R(T) = O\left( dC + \sum_{k=d+1}^{K} \frac{\log(T)\log(KT)}{\Delta_k} + \frac{dK\log(1/\Delta)\log(K/\Delta)}{\Delta} \right).$$

**Remark 9.** *Notice that our DS-BARBAT algorithm can efficiently compute the sampling probability $p_m$, while FTRL-based methods [25, 15, 16, 17] need to solve a complicated convex optimization problem in each round, which is rather expensive. The time comparison in Table 3 in Appendix C.3 demonstrates that our DS-BARBAT algorithm is significantly faster than these FTRL-based methods.*

**Algorithm 5** DS-BARBAT: d-Set-BARBAT

1: **Initialization:** Set the initial round $T_0 \leftarrow 0$, $\Delta_k^0 \leftarrow 1$, and $r_k^0 \leftarrow 0$ for all $k \in [K]$.
2: **for** epochs $m = 1, 2, \cdots$ **do**
3:     Set $\zeta_m \leftarrow (m+4)2^{2(m+4)}\ln(K)$, and $\delta_m \leftarrow 1/(K\zeta_m)$
4:     Set $\lambda_m \leftarrow 2^8 \ln(4K/\delta_m)$ and $\beta_m \leftarrow \delta_m/K$.
5:     Set $n_k^m \leftarrow \lambda_m(\Delta_k^{m-1})^{-2}$ for all arms $k \in [K]$.
6:     Set $N_m \leftarrow \lceil K\lambda_m 2^{2(m-1)}/d \rceil$ and $T_m \leftarrow T_{m-1} + N_m$.
7:     Set the arm subsets $\mathcal{K}_m \leftarrow \arg\max_{k \in [K]} r_k^{m-1}$ with $|\mathcal{K}_m| = d$.
8:     Set $\widetilde{n}_k^m = \begin{cases} n_k^m & k \notin \mathcal{K}_m \\ N_m - \sum_{k \notin \mathcal{K}_m} n_k^m/d & k \in \mathcal{K}_m \end{cases}$.
9:     **for** $t = T_{m-1} + 1$ to $T_m$ **do**
10:         Choose action $\mathbf{x}_t \sim p_m$ where $p_m(k) = \widetilde{n}_k^m/N_m$.
11:         For each arm $k$ with $\mathbf{x}_t(k) = 1$, observe $\widetilde{r}_{t,k}$ and update the total reward $S_k^m \leftarrow S_k^m + \widetilde{r}_{t,k}$.
12:     **end for**
13:     Set $r_k^m \leftarrow \min\{S_k^m/\widetilde{n}_k^m, 1\}$.
14:     Set $r_*^m \leftarrow \top_d\left(\left\{r_k^m - \sqrt{\frac{4\ln(4/\beta_m)}{\widetilde{n}_k^m}}\right\}_{k \in [K]}\right)$, where $\top_d(\mathcal{A})$ returns the $d$-th largest value in $\mathcal{A}$.
15:     Set $\Delta_k^m \leftarrow \max\{2^{-m}, r_*^m - r_k^m\}$.
16: **end for**

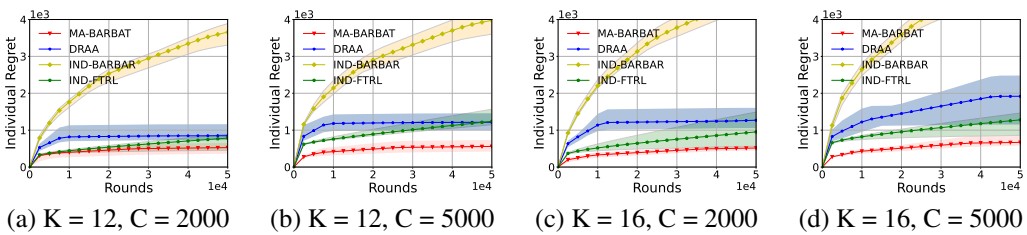

(a) K = 12, C = 2000    (b) K = 12, C = 5000    (c) K = 16, C = 2000    (d) K = 16, C = 5000

Figure 1: Comparison between MA-BARBAT, DRAA, IND-BARBAR and IND-FTRL in cooperative multi-agent multi-armed bandits.

## 5 Experiments

In this section, we present numerical results to demonstrate the robustness and effectiveness of our framework. All experiments are implemented in Python 3.11 and conducted on a Windows laptop equipped with 16 GB of memory and an Intel i7-13700H processor. For all experiments, we run 50 independent trials, report the average results and use shaded regions to represent standard deviations. The detailed experimental settings and time comparisons are deferred to Appendix C.

### 5.1 Cooperative Multi-agent Multi-Armed Bandits

We compare our framework with DRAA [12], IND-BARBAR [1], and IND-FTRL [4] in a multi-agent setting with $V = 10$. Here, IND-BARBAR and IND-FTRL serve as non-cooperative baselines, where each agent runs the algorithm locally without any communication. We do not compare with [32] since the baseline DRAA is an improved version of their method. We present the experimental results in Figure 1, which shows that our BARBAT framework outperforms all baseline methods, reflecting the advantage of collaboration and verifying our theoretical analysis. In addition, when $C$ is increased, the increase in individual regret of our framework is the smallest, which shows that the robustness of our framework is stronger than that of all baseline methods.

### 5.2 Strongly Observable Graph Bandits

We compare our SOG-BARBAT algorithm with Shannon-FTRL [13] and Tsallis-FTRL [14] using two directed graphs as illustrated in Figure 4 in Appendix C.2. We do not include the Elise [10] algorithm in the experiment since it only works for undirected graphs. The experimental results

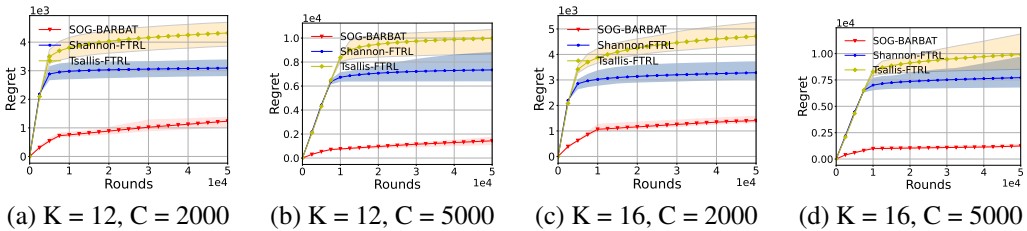

Figure 2: Comparison between SOG-BARBAT, Shannon-FTRL, and Tsallis-FTRL in strongly observable graph bandits.

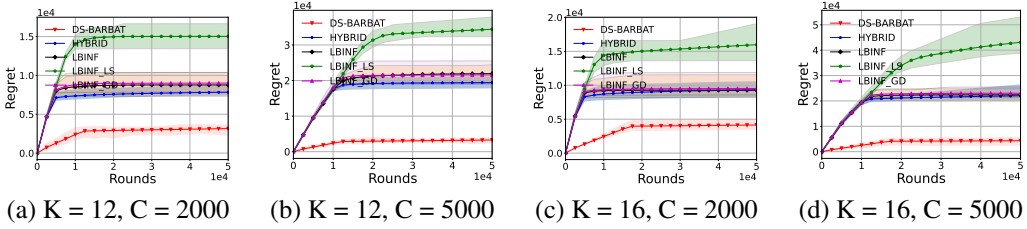

Figure 3: Comparison between DS-BARBAT, HYBRID, LBINF, LBINF_LS and LBINF_GD in $d$-set semi-bandits.

are presented in Figure 2, showing that our SOG-BARBAT algorithm significantly outperforms all baseline methods and verifying the robustness of our method.

### 5.3 $d$-Set Semi-bandits

In $d$-set semi-bandits, we set $d = 3$ for $K = 12$, and $d = 4$ for $K = 16$. We compare our framework with HYBRID [15], LBINF [16], LBINF_LS [17] and LBINF_GD [17]. As shown in Figure 3, our DS-BARBAT algorithm significantly outperforms all baseline methods.

## 6  Conclusion

Building on the BARBAR algorithm [1], we propose the BARBAT framework by elaborately designing the epoch lengths $N_m$ and the parameters $\delta_m$. We further extend BARBAT to various corrupted bandit settings, including multi-agent bandits, batched bandits, strongly observable graph bandits and $d$-set semi-bandits. Our theoretical analysis shows that these algorithms achieve near-optimal regret bounds for adversarially corrupted environments. Compared with FTRL-based approaches, BARBAT and its extensions are more computationally efficient and do not rely on the assumption of a unique optimal action. Moreover, since FTRL-based methods are difficult to parallelize, they are unsuitable for multi-agent and batched settings. These advantages make the BARBAT framework an attractive alternative to FTRL-based algorithms across a wide range of corrupted bandit problems. One future direction is to improve the $\log^2(T)$ dependence in our regret bound to $\log(T)$. Investigating extensions in other bandit settings such as linear bandits is also an interesting future direction.

## Acknowledgement

This work was supported by Natural Science Foundation of China (No. 62306116), and in part by the Key Program of National Natural Science Foundation of China (No. 62432007).

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

# Contents

# A OODS algorithm for strongly observable graph bandits

In this section, we present the OODS algorithm for strongly observable graph bandits. For any graph $G = (V, E)$, the time complexity of the algorithm is $O(|V|^2|E|)$.

---
**Algorithm 6** OODS: Obtain an out-domination Set

---
1: **Input:** a graph $G$.
2: **Output:** an out-domination set $D$.
3: **while** the graph $G$ is not empty **do**
4:    **if** the graph $G$ is an acyclic graph **then**
5:       Find all no-root vertices and add them to the set $D$.
6:       Remove all no-root vertices and their out-degree neighbors from the graph $G$.
7:    **else**
8:       Select the vertex with the largest out-degree in the graph $G$ and add it to the set $D$.
9:       Remove the vertex and its out-degree neighbors from the graph $G$.
10:   **end if**
11: **end while**

---

A "no-root vertex" is defined as a vertex with no incoming edges other than its self-loop. To ensure that the rewards for no-root vertices are observed, OODS pulls these vertices directly. However, in some graphs, there may be no such vertices. In such cases, OODS applies a greedy strategy: it selects the vertex with the highest out-degree and adds it to $D$. The procedure proceeds as follows: if a cycle is detected in $G$, the greedy strategy is employed; otherwise, all no-root vertices are added to $D$. The vertices in $D$ and their out-degree neighbors are then removed from the graph $G$, and the process repeats until $G$ is empty, ensuring that $D$ forms an out-domination set covering $G$.

At a high level, the OODS algorithm finds an out-domination set $D^m$ such that pulling the arms $k \in D^m$ allows us to observe the rewards of all arms. The loop iteratively removes arms once they meet the observation requirement. After each removal, the independence number of the remaining graph may decrease, so the required dominating set becomes smaller—thereby reducing the number of pulls allocated to suboptimal arms.

# B Proof Details

## B.1 Proof of Theorem 1

### B.1.1 Notations

We define $C_k^m$ as the sum of corruptions to arm $k$ in epoch $m$, and let $C_m \triangleq \max_{k \in [K]} C_k^m$.

### B.1.2 Lemmas for Proving Theorem 1

**Lemma 1.** *For the BARBAT algorithm time horizon $T$, the number of epochs $M$ is at most $\log_2(T)$. In the $m$-th epoch, the selected arm $k_m$ must satisfy $\Delta_{k_m}^{m-1} = 2^{-(m-1)}$.*

*Proof.* The length of epoch $m$ is given by $N_m = \lceil K\lambda_m 2^{2(m-1)} \rceil \geq 2^{2(m-1)}$. From the lower bound of $N_m$, we can complete the first statement. Since $\Delta_k^{m-1} \leftarrow \max\{2^{-(m-1)}, r_*^m - r_k^m\}$, there exists at least one arm that satisfies $\Delta_k^{m-1} = 2^{-(m-1)}$ and all arms satisfy $\Delta_k^{m-1} \leq 2^{-(m-1)}$. Since $r_{k_m}^{m-1} > r_*^{m-1}$, the equality $\Delta_{k_m}^{m-1} = 2^{-(m-1)}$ must hold. $\qquad\square$

**Lemma 2.** *For any epoch $m$, the length $N_m$ satisfies $N_m \geq \sum_{k \in [K]} n_k^m$.*

*Proof.* Since $\Delta_k^m = \max\{2^{-m}, r_*^m - r_k^m\} \geq 2^{-m}$, we can get $n_k^m = \lambda_m (\Delta_k^{m-1})^{-2} \leq \lambda_m 2^{2(m-1)}$. Therefore, we have $\sum_{k \in [K]} n_k^m \leq K\lambda_m 2^{2(m-1)} \leq N_m$. $\qquad\square$

**Lemma 3.** *For epoch $s$ and $m$ with $1 \leq s \leq m$, the following inequality holds:*

$$\frac{\lambda_m}{\lambda_s} \leq \left(\frac{7}{6}\right)^{m-s}.$$

*Proof.* We first show that the function $f(x) = \frac{(x+1.7)\ln(4)+\ln(x)}{(7/6)^x}$ is strictly decreasing for $x \geq 5$. Notice that the derivative function

$$f'(x) = \frac{(\ln(4)+1/x) - ((x+1.7)\ln(4)+\ln(x))\ln\left(\frac{7}{6}\right)}{\left(\frac{7}{6}\right)^x}$$

is monotonically decreasing and $f'(5) < 0$, thus we have $f'(x) < 0$ for $x \geq 5$, which indicates that $f(x)$ is strictly decreasing. Since $K \geq 2$, we can get

$$\begin{aligned}
\frac{\lambda_m}{\lambda_s} &= \frac{\ln(4K^2 \ln(K)(m+4)2^{2(m+4)})}{\ln(4K^2 \ln(K)(s+4)2^{2(s+4)})} \\
&= \frac{\ln(4K^2 \ln(K)) + \ln(m+4) + (m+4)\ln(4)}{\ln(4K^2 \ln(K)) + \ln(s+4) + (s+4)\ln(4)} \\
&< \frac{1.7 \cdot \ln(4) + \ln(m+4) + (m+4)\ln(4)}{1.7 \cdot \ln(4) + \ln(s+4) + (s+4)\ln(4)} \\
&= \frac{f(m+4)}{f(s+4)}\left(\frac{7}{6}\right)^{m-s} < \left(\frac{7}{6}\right)^{m-s}
\end{aligned}$$

where we use the monotonicity of $f(x)$ and the fact that $K \geq 2$. $\qquad\square$

**Lemma 4.** *For any epoch $m$, the following inequality holds:*

$$\sum_{s=1}^m \lambda_s \leq 2^8(m^2 + m(10 + 3\ln(K))).$$

*Proof.* Given the function $f(x) = 2x(1 - \ln(2)) - \ln(x+4) + 7 - 8\ln(2)$. Notice that the derivative function as $f'(x) = 2 - 2\ln(2) - \frac{1}{x+4} > 0$ for all $x \geq 1$, which means that $f(x)$ is strictly increasing

for $x \geq 1$. Since $f(x) \geq f(1) \geq 0$, we can get the inequality as $2x + 7 \geq 2(x+4)\ln(2) + \ln(x+4)$ for all $x \geq 1$, then have

$$
\begin{aligned}
\sum_{s=1}^{m} \lambda_s &= \sum_{s=1}^{m} 2^8 (\ln(4K^2 \ln(K)(s+4)2^{2(s+4)})) \\
&= \sum_{s=1}^{m} 2^8 (2(s+4)\ln(2) + \ln(s+4) + \ln(4K^2 \ln(K)) \\
&\leq \sum_{s=1}^{m} 2^8 (2s + 7 + \ln(4K^3)) \\
&= 2^8 (m^2 + 8m + m\ln(4K^3)) \\
&\leq 2^8 (m^2 + m(10 + 3\ln(K))).
\end{aligned}
$$

$\square$

**Lemma 5.** *For any fixed $k, m$ and $\beta_m$, Algorithm 1 satisfies*

$$
\Pr\left[|r_k^m - \mu_k| \geq \sqrt{\frac{4\ln(4/\beta_m)}{\widetilde{n}_k^m}} + \frac{2C_m}{N_m}\right] \leq \beta_m.
$$

*Proof.* The proof of Lemma 5 mainly follows the proof of Lemma 4 in [1]. We only consider the randomness in epoch $m$ and condition on all random variables before epoch $m$. In epoch $m$, the agent pulls arm $k$ with probability $p_m(k) = \widetilde{n}_k^m / N_m$.

For ease of analysis, we set an indicator variable $Y_k^t$, which determines whether the agent updates the corrupted reward $\widetilde{r}_{I_t}$ into the total reward $S_{I_t}^m$ at step $t$. We define the corruption at step $t$ on arm $k$ as $C_k^t := \widetilde{r}_k^t - r_k^t$. Let $E_m := [T_{m-1} + 1, ..., T_m]$ represent the $N_m$ time-steps for epoch $m$. Since $r_k^m = \min\{S_k^m / \widetilde{n}_k^m, 1\}$, we can obtain

$$
r_k^m \leq \frac{S_k^m}{\widetilde{n}_k^m} = \frac{1}{\widetilde{n}_k^m} \sum_{t \in E_m} Y_k^t (r_k^t + C_k^t).
$$

We can divide the sum into two components:

$$
A_k^m = \sum_{t \in E_m} Y_k^t r_k^t, \quad B_k^m = \sum_{t \in E_m} Y_k^t C_k^t.
$$

For the previous component $A_k^m$, notice that $r_k^t$ is independently drawn from a $[0,1]$-valued distribution with mean $\mu_k$, and $Y_k^t$ is independently drawn from a Bernoulli distribution with mean $q_k^m := \widetilde{n}_k^m / N_m$. Therefore, we have

$$
\mathbb{E}[A_k^m] = N_m q_k^m \mathbb{E}[r_k^t] = \widetilde{n}_k^m \mu_k \leq \widetilde{n}_k^m.
$$

By applying the Chernoff-Hoeffding inequality (Theorem 1.1 in [33]), we can get

$$
\Pr\left[\left|\frac{A_k^m}{\widetilde{n}_k^m} - \mu_k\right| \geq \sqrt{\frac{3\ln(4/\beta_m)}{\widetilde{n}_k^m}}\right] \leq \frac{\beta_m}{2}. \tag{2}
$$

For analyzing the latter component $B_k^m$, we define a martingale difference sequence $X_1, ..., X_T$, where $X_t = (Y_k^t - q_k^m)C_k^t$ for all $t$, with respect to the historical information $\{\mathcal{F}\}_{t=1}^{T}$. Since the corruption $C_k^t$ is deterministic when conditioned on $\mathcal{F}_{t-1}$ and since $\mathbb{E}[Y_k^t | \mathcal{F}_{t-1}] = q_k^m$, we can get

$$
\mathbb{E}[X_t | \mathcal{F}_{t-1}] = \mathbb{E}[Y_k^t - q_k^m | \mathcal{F}_{t-1}]C_k^t = 0.
$$

Additionally, we have $|X_t| \leq 1$ for all $t$, and the predictable quadratic variation of this martingale can be bounded as follows:

$$
\sum_{t \in E_m} \mathbb{E}[(X_t)^2 | \mathcal{F}_{t-1}] \leq \sum_{t \in E_m} |C_k^t| \mathrm{Var}(Y_k^t) \leq q_k^m \sum_{t \in E_m} |C_k^t|.
$$

By Freedman concentration inequality (Theorem 1 in [34]), with probability at least $1 - \beta_m/4$ we have

$$B_k^m \leq q_k^m \sum_{t \in E_m} C_k^t + \sum_{t \in E_m} \mathbb{E}[(X_t)^2 | \mathcal{F}_{t-1}] + \ln(4/\beta_m) \leq 2q_k^m \sum_{t \in E_m} |C_k^t| + \ln(4/\beta_m).$$

Since $q_k^m = \widetilde{n}_k^m/N_m$, $\sum_{t \in E_m} |C_k^t| \leq C_m$ and $n_k^m \geq \lambda_m \geq 16 \ln(4/\beta_m)$, with probability at least $1 - \beta_m/4$ the following inequality holds:

$$\frac{B_k^m}{\widetilde{n}_k^m} \leq \sqrt{\frac{\ln(4/\beta_m)}{16\widetilde{n}_k^m}} + \frac{2C_m}{N_m}.$$

Similarly, with probability at least $1 - \beta_m/4$ the following inequality holds:

$$-\frac{B_k^m}{\widetilde{n}_k^m} \leq \sqrt{\frac{\ln(4/\beta_m)}{16\widetilde{n}_k^m}} + \frac{2C_m}{N_m}.$$

Thus we have

$$\Pr\left[\left|\frac{B_k^m}{\widetilde{n}_k^m}\right| \geq \sqrt{\frac{\ln(4/\beta_m)}{16\widetilde{n}_k^m}} + \frac{2C_m}{N_m}\right] \leq \frac{\beta_m}{2}. \tag{3}$$

Combine Eq. (2) and Eq. (3), we complete the proof. $\qquad\square$

We define an event $\mathcal{E}_m$ for epoch $m$ as follows:

$$\mathcal{E}_m \triangleq \left\{ \forall k : |r_k^m - \mu_k| \leq \sqrt{\frac{4\ln(4/\beta_m)}{\widetilde{n}_k^m}} + \frac{2C_m}{N_m} \right\}.$$

Then we can establish a lower bound on the probability of the event $\mathcal{E}_m$ occurring by the following lemma.

**Lemma 6.** *For any epoch $m$, event $\mathcal{E}_m$ holds with probability at least $1 - \delta_m$. We also have $1/\delta_m \geq N_m$.*

*Proof.* By Lemma 5, we can get

$$\Pr\left[|r_k^m - \mu_k| \leq \sqrt{\frac{4\ln(4/\beta_m)}{\widetilde{n}_k^m}} + \frac{2C_m}{N_m}\right] \leq \beta_m = \frac{\delta_m}{K}.$$

A union bound over the $K$ arms indicates that the success probability of event $\mathcal{E}_m$ is at least $1 - \delta_m$.

Since $m \geq 1$ and $K \geq 2$, then we can get

$$\begin{aligned}
N_m &= K2^{2(m-1)}\lambda_m \\
&= K2^{2(m+3)} \ln((m+4)2^{2(m+5)}K^2 \ln(K)) \\
&\leq K2^{2(m+3)}(\ln(m+4) + 2(m+5)\ln(2) + \ln(K^3)) \\
&\leq K2^{2(m+3)}((m+4)\ln(K) + 2(m+5)\ln(K) + 3\ln(K)) \\
&\leq K2^{2(m+3)}(4m+16)\ln(K) \\
&= K2^{2(m+4)}((m+4)\ln(K)) = 1/\delta_m.
\end{aligned}$$

$\qquad\square$

Now we can define the offset level $D_m$ for each epoch $m$:

$$D_m = \begin{cases} 2C_m & \text{when } \mathcal{E}_m \text{ occurs} \\ N_m & \text{when } \mathcal{E}_m \text{ does not occur} \end{cases}.$$

Since we always have $|r_k^m - \mu_k| \leq 1$, thus the following inequality always holds, regardless of whether event $\mathcal{E}_m$ happens:

$$|r_k^m - \mu_k| \leq \sqrt{\frac{4\ln(4/\beta_m)}{\widetilde{n}_k^m}} + \frac{D_m}{N_m}.$$

By the definition of $D_m$, we have

$$\Pr[D_m = 2C_m] \geq 1 - \delta_m \quad \text{and} \quad \Pr[D_m = N_m] \leq \delta_m.$$

Next, we will bound the estimated gap $\Delta_k^m$. To start, we define the discounted offset rate as

$$\rho_m := \sum_{s=1}^m \frac{D_s}{8^{m-s} N_s}.$$

Then we have the following lemma.

**Lemma 7.** *For all epochs $m$ and arms $k$, we can have*

$$\frac{4\Delta_k}{7} - \frac{3}{2}2^{-m} - 6\rho_m \leq \Delta_k^m \leq \frac{8\Delta_k}{7} + 2^{-(m-1)} + 2\rho_m.$$

*Proof.* Since $|r_k^m - \mu_k| \leq \sqrt{\frac{4\ln(4/\beta_m)}{\widetilde{n}_k^m}} + \frac{D_m}{N_m}$, we have

$$-\frac{D_m}{N_m} - \sqrt{\frac{4\ln(4/\beta_m)}{\widetilde{n}_k^m}} \leq r_k^m - \mu_k \leq \frac{D_m}{N_m} + \sqrt{\frac{4\ln(4/\beta_m)}{\widetilde{n}_k^m}}.$$

Additionally, since

$$r_*^m \leq \max_k \left\{ \mu_k + \frac{D_m}{N_m} + \sqrt{\frac{4\ln(4/\beta_m)}{\widetilde{n}_k^m}} - \sqrt{\frac{4\ln(4/\beta_m)}{\widetilde{n}_k^m}} \right\} \leq \mu_{k^*} + \frac{D_m}{N_m},$$

$$r_*^m = \max_k \left\{ r_k^m - \sqrt{\frac{4\ln(4/\beta_m)}{\widetilde{n}_k^m}} \right\} \geq r_{k^*}^m - \sqrt{\frac{4\ln(4/\beta_m)}{\widetilde{n}_{k^*}^m}} \geq \mu_{k^*} - 2\sqrt{\frac{4\ln(4/\beta_m)}{\widetilde{n}_{k^*}^m}} - \frac{D_m}{N_m},$$

we can get

$$-\frac{D_m}{N_m} - 2\sqrt{\frac{4\ln(4/\beta_m)}{\widetilde{n}_{k^*}^m}} \leq r_*^m - \mu_{k^*} \leq \frac{D_m}{N_m}.$$

According to Algorithm 1 and Lemma 2, we have $\widetilde{n}_k^m \geq n_k^m$ for all arms $k$. Then we have the following inequality for all $k \in [K]$:

$$\sqrt{\frac{4\ln(4/\beta_m)}{\widetilde{n}_k^m}} \leq \sqrt{\frac{4\ln(4/\beta_m)}{n_k^m}} = \frac{\Delta_k^{m-1}}{8}.$$

We now establish the upper bound for $\Delta_k^m$ by induction on $m$.

For the base case $m = 1$, the statement is trivial as $\Delta_k^1 = 1$ for all $k \in [K]$.

Assuming the statement is true for the case of $m - 1$, we have

$$\begin{aligned}
\Delta_k^m = r_*^m - r_k^m &= (r_*^m - \mu_{k^*}) + (\mu_{k^*} - \mu_k) + (\mu_k - r_k^m) \\
&\leq \frac{D_m}{N_m} + \Delta_k + \frac{D_m}{N_m} + \frac{1}{8}\Delta_k^{m-1} \\
&\leq \frac{2D_m}{N_m} + \Delta_k + \frac{1}{8}\left( \frac{8\Delta_k}{7} + 2^{-(m-2)} + 2\rho_{m-1} \right) \\
&\leq \frac{8\Delta_k}{7} + 2^{-(m-1)} + 2\rho_m,
\end{aligned}$$

Where the second inequality follows from the induction hypothesis.

Next, we provide the lower bound of $\Delta_k^m$. We can get

$$\Delta_k^m = r_*^m - r_k^m = (r_*^m - \mu_{k^*}) + (\mu_{k^*} - \mu_k) + (\mu_k - r_k^m)$$

$$\geq -\frac{D_m}{N_m} - \frac{1}{4}\Delta_{k^*}^{m-1} + \Delta_k - \frac{D_m}{N_m} - \frac{1}{8}\Delta_k^{m-1}$$

$$\geq -\frac{2D_m}{N_m} + \Delta_k - \frac{3}{8}\left(\frac{8\Delta_k}{7} + 2^{-(m-2)} + 2\rho_{m-1}\right)$$

$$\geq \frac{4}{7}\Delta_k - \frac{3}{2}2^{-m} - 6\rho_m.$$

where the third inequality comes from the upper bound of $\Delta_k^{m-1}$. $\qquad\square$

### B.1.3 Proof for Theorem 1

We first define the regret $R_k^m$ generated by arm $k$ in epoch $m$ as

$$R_k^m \triangleq \Delta_k \widetilde{n}_k^m = \begin{cases} \Delta_k n_k^m & k \neq k_m \\ \Delta_k \widetilde{n}_k^m & k = k_m \end{cases}.$$

Then we analyze the regret in following three cases:

**Case 1:** $0 < \Delta_k \leq 64\rho_{m-1}$.
If $k \neq k_m$, then we have

$$R_k^m = \Delta_k n_k^m \leq 64\rho_{m-1} n_k^m.$$

If $k = k_m$, then we have

$$R_k^m = \Delta_k \widetilde{n}_{k_m}^m \leq 64\rho_{m-1} N_m.$$

**Case 2:** $\Delta_k \leq 8 \cdot 2^{-m}$ and $\rho_{m-1} \leq \frac{\Delta_k}{64}$.
If $k \neq k_m$, then we have

$$R_k^m = n_k^m \Delta_k = \lambda_m (\Delta_k^{m-1})^{-2} \Delta_k \leq \lambda_m 2^{2(m-1)} \Delta_k \leq \frac{16\lambda_m}{\Delta_k}.$$

If $k = k_m$, since $\Delta_{k_m}^{m-1} = 2^{-(m-1)}$, we can get:

$$R_k^m = \widetilde{n}_{k_m}^m \Delta_{k_m} \leq N_m \Delta_{k_m} = \lceil K\lambda_m 2^{2(m-1)} \rceil \Delta_{k_m} \leq \frac{16K\lambda_m}{\Delta_{k_m}} + \Delta_{k_m} \leq \frac{16K\lambda_m}{\Delta} + 1.$$

**Case 3:** $\Delta_k > 8 \cdot 2^{-m}$ and $\rho_{m-1} \leq \frac{\Delta_k}{64}$.
By Lemma 7 we have

$$\Delta_k^{m-1} \geq \frac{4}{7}\Delta_k - \frac{3}{2}2^{-m} - \frac{6}{64}\Delta_k \geq \Delta_k\left(\frac{4}{7} - \frac{3}{16} - \frac{6}{64}\right) \geq 0.29\Delta_k.$$

In this case, it is impossible that $k = k_m$ because $\Delta_{k_m}^{m-1} = 2^{-(m-1)} < 0.29 \cdot 8 \cdot 2^{-m} < 0.29\Delta_k$.
So we can obtain

$$R_k^m = n_k^m \Delta_k = \lambda_m (\Delta_k^{m-1})^{-2} \Delta_k \leq \frac{\lambda_m}{0.29^2 \Delta_k} \leq \frac{16\lambda_m}{\Delta_k}.$$

We define $\mathcal{A}^m \triangleq \{k \in [K] \,\big|\, 0 < \Delta_k \leq 64\rho_{m-1}\}$ for epoch $m$. By combining all three cases, we can upper bound the regret as

$$
\begin{aligned}
R(T) &= \sum_{m=1}^{M} \left( \sum_{k \in \mathcal{A}^m} R_k^m + \sum_{k \notin \mathcal{A}^m} R_k^m \right) \\
&\leq \sum_{m=1}^{M} \left( 64\rho_{m-1}N_m + \sum_{k \in \mathcal{A}^m, k \neq k_m} 64\rho_{m-1}n_k^m + \sum_{k \notin \mathcal{A}^m, \Delta_k > 0} \frac{16\lambda_m}{\Delta_k} \right. \\
&\quad \left. + \left( \frac{16K\lambda_m}{\Delta} + 1 \right) \mathbb{I}(0 < \Delta_{k_m} \leq 8 \cdot 2^{-m}) \right) \\
&\leq \sum_{m=1}^{M} \left( 64\rho_{m-1}N_m + \sum_{\Delta_k > 0} 64\rho_{m-1}n_k^m + \sum_{\Delta_k > 0} \frac{16\lambda_m}{\Delta_k} \right. \\
&\quad \left. + \left( \frac{16K\lambda_m}{\Delta} + 1 \right) \mathbb{I}\left(m \leq \log_2\left(8/\Delta\right)\right) \right) \\
&\leq \sum_{m=1}^{M} \left( 128\rho_{m-1}N_m + \sum_{\Delta_k > 0} \frac{16\lambda_m}{\Delta_k} \right) + \sum_{m=1}^{\log_2(8/\Delta)} \left( \frac{16K\lambda_m}{\Delta} + 1 \right)
\end{aligned} \tag{4}
$$

where the last inequality uses the fact that $\sum_{\Delta_k > 0} n_k^m \leq N_m$. Notice that we can bound the expectation of the offset level as

$$
\mathbb{E}[D_m] = 2(1 - \delta_m)C_m + \delta_m N_m \leq 2C_m + 1
$$

and we can bound $\sum_{m=1}^{M} \rho_{m-1}N_m$ as

$$
\begin{aligned}
\sum_{m=1}^{M} \rho_{m-1}N_m &\leq \sum_{m=1}^{M} \left( \sum_{s=1}^{m-1} \frac{D_s}{8^{m-1-s}N_s} \right) N_m \\
&\leq 4 \sum_{m=1}^{M} \left( \sum_{s=1}^{m-1} \frac{(4^{m-1-s} + 1)\lambda_m}{8^{m-1-s}\lambda_s} D_s \right) \\
&= 4 \sum_{m=1}^{M} \left( \sum_{s=1}^{m-1} ((7/12)^{m-1-s} + (7/48)^{m-1-s})D_s \right) \\
&= 4 \sum_{s=1}^{M-1} D_s \sum_{m=s+1}^{M} (7/12)^{m-1-s} + (7/48)^{m-1-s} \\
&\leq 4 \left( \sum_{m=1}^{M-1} D_m \right) \sum_{j=0}^{\infty} (7/12)^j + (7/48)^j \leq 11 \sum_{m=1}^{M-1} D_m.
\end{aligned} \tag{5}
$$

Combining Eq. (4) and Eq. (5), by Lemma 4, we can get

$$
\begin{aligned}
R(T) &\leq \sum_{m=1}^{M-1} 1440\mathbb{E}[D_m] + \sum_{m=1}^{M}\sum_{\Delta_k>0}\frac{16\lambda_m}{\Delta_k} + \sum_{m=1}^{\log_2(8/\Delta)}\left(\frac{16K\lambda_m}{\Delta}+1\right) \\
&\leq \sum_{m=1}^{M-1} 1440\mathbb{E}[D_m] + \sum_{\Delta_k>0}\frac{2^{12}(\log^2(T)+3\log(T)\log(30K))}{\Delta_k} \\
&\quad + \frac{2^{12}K(\log^2(8/\Delta)+3\log(8/\Delta)\log(30K)))}{\Delta} + \log(8/\Delta) \\
&\leq \sum_{m=1}^{M-1} 2880C_m + 1440 + \sum_{\Delta_k>0}\frac{2^{14}\log(T)\log(30KT)}{\Delta_k} \\
&\quad + \frac{2^{14}K\log(8/\Delta)\log(240K/\Delta)}{\Delta} + \log(8/\Delta) \\
&= O\left(C + \sum_{\Delta_k>0}\frac{\log(T)\log(KT)}{\Delta_k} + \frac{K\log(1/\Delta)\log(K/\Delta)}{\Delta}\right).
\end{aligned}
$$

## B.2 Proof of Theorem 2

### B.2.1 Notations

We define $C_k^m$ as the sum of corruptions to arm $k$ in epoch $m$ for all agents $v \in [V]$, and let $C_m \triangleq \max_{k\in[K]} C_k^m$.

### B.2.2 Lemmas for Proving Theorem 2

**Lemma 8.** *For BARBAT with time horizon $T$, the number of epochs $M$ is at most $\log(VT)$. In the $m$-th epoch, the selected arm $k_m$ must satisfy $\Delta_{k_m}^{m-1} = 2^{-(m-1)}$ for all agents $v \in [V]$.*

*Proof.* The length of epoch $m$ is given by $N_m = \lceil K\lambda_m 2^{2(m-1)}\rceil \geq 2^{2(m-1)}/V$. From the lower bound of $N_m$, we can complete the first statement. Since $\Delta_k^{m-1} \leftarrow \max\{2^{-(m-1)}, r_*^m - r_k^m\}$, there exists at least one arm that satisfies $\Delta_k^{m-1} = 2^{-(m-1)}$ and all arms satisfy $\Delta_k^{m-1} \leq 2^{-(m-1)}$. Since $r_{k_m}^{m-1} > r_*^{m-1}$, the equality $\Delta_{k_m}^{m-1} = 2^{-(m-1)}$ must hold. $\qquad\square$

Since the length $N_m$ still satisfies $N_m = \lceil K\lambda_m 2^{2(m-1)}\rceil \geq \sum_{k\in[K]} n_k^m$, Lemma 2 still holds for this setting.

**Lemma 9.** *For epoch $s$ and $m$ with $1 \leq s \leq m$, the following inequality holds:*

$$
\frac{\lambda_m}{\lambda_s} \leq \left(\frac{7}{6}\right)^{m-s}.
$$

*Proof.* We first show that the function $f(x) = \frac{(x+1.7)\ln(4)+\ln(x)}{(7/6)^x}$ is strictly decreasing for $x \geq 5$. Notice that the derivative function

$$
f'(x) = \frac{(\ln(4)+1/x)-((x+1.7)\ln(4)+\ln(x))\ln\left(\frac{7}{6}\right)}{\left(\frac{7}{6}\right)^x}
$$

is monotonically decreasing and $f'(5) < 0$, thus we have $f'(x) < 0$ for $x \geq 5$, which indicates that $f(x)$ is strictly decreasing. Since $K \geq 2$ and $V \geq 1$, we can get

$$
\begin{aligned}
\frac{\lambda_m}{\lambda_s} &= \frac{\ln(4VK^2\ln(VK)(m+4)2^{2(m+4)})}{\ln(4VK^2\ln(VK)(s+4)2^{2(s+4)})} \\
&= \frac{\ln(4VK^2\ln(VK))+\ln(m+4)+(m+4)\ln(4)}{\ln(4VK^2\ln(VK))+\ln(s+4)+(s+4)\ln(4)}
\end{aligned}
$$

$$< \frac{1.7 \cdot \ln(4) + \ln(m+4) + (m+4)\ln(4)}{1.7 \cdot \ln(4) + \ln(s+4) + (s+4)\ln(4)}$$

$$= \frac{f(m+4)}{f(s+4)} \left(\frac{7}{6}\right)^{m-s} < \left(\frac{7}{6}\right)^{m-s}$$

where we use the monotonicity of $f(x)$ and the fact that $K \geq 2$ and $V \geq 1$. $\qquad \square$

**Lemma 10.** *For any epoch $m$, the following inequality holds:*

$$\sum_{s=1}^{m} \lambda_s \leq 2^8(m^2 + m(10 + \ln(VK))).$$

*Proof.* Given the function $f(x) = 2x(1 - \ln(2)) - \ln(x+4) + 7 - 8\ln(2)$. Notice that the derivative function as $f'(x) = 2 - 2\ln(2) - \frac{1}{x+4} > 0$ for all $x \geq 1$, which means that $f(x)$ is strictly increasing for $x \geq 1$. Since $f(x) \geq f(1) \geq 0$, we can get the inequality as $2x + 7 \geq 2(x+4)\ln(2) + \ln(x+4)$ for all $x \geq 1$, then have

$$\sum_{s=1}^{m} \lambda_s = \sum_{s=1}^{m} 2^8(\ln(4VK^2\ln(VK)(s+4)2^{2(s+4)}))$$

$$= \sum_{s=1}^{m} 2^8(2(s+4)\ln(2) + \ln(s+4) + \ln(4VK^2\ln(VK))$$

$$\leq \sum_{s=1}^{m} 2^8(2s + 7 + \ln(4V^2K^3))$$

$$= 2^8(m^2 + 8m + m\ln(4V^2K^3))$$

$$\leq 2^8(m^2 + m(10 + 3\ln(VK))).$$

$\qquad \square$

**Lemma 11.** *For any fixed $k, m$ and any $\beta_m \geq 4e^{-V\lambda_m/16}$, Algorithm 2 satisfies*

$$\Pr\left[|r_k^m - \mu_k| \geq \sqrt{\frac{4\ln(4/\beta_m)}{V\widetilde{n}_k^m}} + \frac{2C_m}{VN_m}\right] \leq \beta_m.$$

*Proof.* Since in each epoch $m$, all agents have the same probability $p_m(k)$ of pulling each arm $k$. Using the method as Lemma 5, we set an indicator variable $Y_{v,k}^t$, which determines whether the agent $v$ updates the corrupted reward $\widetilde{r}_{v,I_t}$ into the total reward $S_{v,I_t}^m$ at step $t$. We define the corruption at step $t$ on arm $k$ for agent $v$ as $C_{v,k}^t := \widetilde{r}_{v,k}^t - r_{v,k}^t$. Let $E_m := [T_{m-1} + 1, ..., T_m]$ represent the $N_m$ time-steps for epoch $m$. Since $r_k^m = \min\{\sum_{v\in[V]} S_{v,k}^m/(V\widetilde{n}_k^m), 1\}$, we can obtain

$$r_k^m \leq \sum_{v\in[V]} \frac{S_{v,k}^m}{V\widetilde{n}_k^m} = \frac{1}{V\widetilde{n}_k^m} \sum_{t\in E_m} \sum_{v\in[V]} Y_{v,k}^t(r_{v,k}^t + C_{v,k}^t).$$

We can divide the sum to two components:

$$A_k^m = \sum_{t\in E_m} \sum_{v\in[V]} Y_{v,k}^t r_{v,k}^t, \quad B_k^m = \sum_{t\in E_m} \sum_{v\in[V]} Y_{v,k}^t C_{v,k}^t.$$

For the previous component $A_k^m$, notice that $r_k^t$ is dependently drawn from an unknown distribution with mean $\mu_k$, and $Y_{v,k}^t$ is dependently drawn from a Bernoullidistribution with mean $q_k^m := \widetilde{n}_k^m/N_m$. Therefore, we have

$$\mathbb{E}[A_k^m] = VN_m q_k^m \mathbb{E}[r_k^t] = V\widetilde{n}_k^m \mu_k \leq V\widetilde{n}_k^m.$$

By applying the Chernoff-Hoeffding inequality (Theorem 1.1 in [33]), we can get

$$\Pr\left[\left|\frac{A_k^m}{V\widetilde{n}_k^m} - \mu_k\right| \geq \sqrt{\frac{3\ln(4/\beta_m)}{V\widetilde{n}_k^m}}\right] \leq \frac{\beta_m}{2}. \tag{6}$$

For the latter component $B_k^m$, we need to define a martingale difference sequence $X_i^1, ..., X_i^T$, where $X_i^t = (Y_{v,k}^t - q_k^m)C_{v,k}^t$ for all $t$, with respect to the historical information $\{\mathcal{F}\}_{t=1}^T$. Since the corruption $C_{v,k}^t$ becomes a deterministic value when conditioned on $\mathcal{F}_{t-1}$ and $\mathbb{E}[Y_{v,k}^t|\mathcal{F}_{t-1}] = q_k^m$, we can get

$$\mathbb{E}[X_i^t|\mathcal{F}_{t-1}] = \mathbb{E}[Y_{v,k}^t - q_k^m|\mathcal{F}_{t-1}]C_{v,k}^t = 0.$$

Additionally, we have $|X_i^t| \leq 1$ for all $t$ and all $v \in [V]$, and the predictable quadratic variation of this martingale can be bounded as follows:

$$\text{Var} = \sum_{t \in E_m} \sum_{v \in [V]} \mathbb{E}[(X_i^t)^2|\mathcal{F}_{t-1}] \leq \sum_{t \in E_m} \sum_{v \in [V]} |C_{v,k}^t|\text{Var}(Y_{v,k}^t) \leq q_k^m \sum_{t \in E_m} \sum_{v \in [V]} |C_{v,k}^t|.$$

By applying the concentration inequality for martingales (Theorem 1 in [34]), with probability at least $1 - \frac{\beta_m}{4}$, we have

$$B_k^m \leq q_k^m \sum_{t \in E_m} \sum_{v \in [V]} C_{v,k}^t + \text{Var} + \ln(4/\beta_m) \leq 2q_k^m \sum_{t \in E_m} \sum_{v \in [V]} |C_{v,k}^t| + \ln(4/\beta_m).$$

Since $q_k^m = \widetilde{n}_k^m/N_m$, $\sum_{t \in E_m} \sum_{v \in [V]} |C_{v,k}^t| \leq C_m$ and $n_k^m \geq \lambda_m \geq 16\ln(4/\beta_m)/V$, with probability at least $1 - \frac{\beta_m}{4}$, we can get the following inequality:

$$\frac{B_k^m}{V\widetilde{n}_k^m} \leq \sqrt{\frac{\ln(4/\beta_m)}{16V\widetilde{n}_k^m}} + \frac{2C_m}{V\widetilde{n}_k^m}.$$

Similarly, $-\frac{B_k^m}{V\widetilde{n}_k^m}$ also meets this bound with probability $1 - \beta/4$. Therefore, we have

$$\Pr\left[\left|\frac{B_k^m}{V\widetilde{n}_k^m}\right| \geq \sqrt{\frac{\ln(4/\beta_m)}{16V\widetilde{n}_k^m}} + \frac{2C_m}{VN_m}\right] \leq \frac{\beta_m}{2}. \tag{7}$$

Combine Eq. 6 and Eq. 7, we complete the proof. $\qquad\square$

We also define an event $\mathcal{E}_m$ for epoch $m$ as follows:

$$\mathcal{E}_m \triangleq \left\{\forall k : |r_k^m - \mu_k| \leq \sqrt{\frac{4\ln(4/\beta_m)}{V\widetilde{n}_k^m}} + \frac{2C_m}{VN_m}\right\}.$$

**Lemma 12.** *For any epoch $m$, event $\mathcal{E}_m$ holds with probability at least $1 - \delta_m$. And after rigorous calculation, we have $1/\delta_m \geq VN_m$.*

*Proof.* By Lemma 11, we can get

$$\Pr\left[|r_k^m - \mu_k| \leq \sqrt{\frac{4\ln(4/\beta_m)}{V\widetilde{n}_k^m}} + \frac{2C_m}{VN_m}\right] \leq 2\beta_m = \frac{\delta_m}{VK}.$$

A union bound over the $V$ agents and the $K$ arms conclude the proof.

Since $m \geq 1$, $V \geq 1$ and $K \geq 2$, then we can get

$$
\begin{aligned}
VN_m &= KV2^{2(m-1)}\lambda_m \\
&= K2^{2(m+3)}\ln((m+4)2^{2(m+5)}VK^2\ln(K)) \\
&\leq K2^{2(m+3)}(\ln(m+4) + 2(m+5)\ln(2) + \ln(VK^3)) \\
&\leq K2^{2(m+4)}((m+4)\ln(VK)) \\
&\leq 1/\delta_m.
\end{aligned}
$$

$\qquad\square$

As mentioned before, for each epoch $m$, to unify the varying bounds depending on the occurrence of event $\mathcal{E}_m$, we also define the offset level

$$D_m = \begin{cases} 2C_m & \text{when } \mathcal{E}_m \text{ occurs} \\ VN_m & \text{when } \mathcal{E}_m \text{ does not occur} \end{cases}.$$

This way we can always guarantee the following inequality:

$$|r_k^m - \mu_k| \leq \sqrt{\frac{4\ln(4/\beta_m)}{V\widetilde{n}_k^m}} + \frac{D_m}{VN_m}.$$

By the definition of $D_m$, we have

$$\Pr[D_m = 2C_m] \geq 1 - \delta_m \quad \text{and} \quad \Pr[D_m = N_m] \leq \delta_m.$$

Next, we will bound the estimated gap $\Delta_k^m$. To start, we define the discounted offset rate as

$$\rho_m := \sum_{s=1}^{m} \frac{D_s}{8^{m-s}VN_s}.$$

Then we have the following lemma.

**Lemma 13.** *For all epochs $m$ and arms $k$, we can have*

$$\frac{4}{7}\Delta_k - \frac{3}{4}2^{-m} - 6\rho_m \leq \Delta_k^m \leq \frac{8\Delta_k}{7} + 2^{-(m-1)} + 2\rho_m.$$

*Proof.* Since $|r_k^m - \mu_k| \leq \sqrt{\frac{4\ln(4/\beta_m)}{V\widetilde{n}_k^m}} + \frac{D_m}{VN_m}$, we have

$$-\frac{D_m}{VN_m} - \sqrt{\frac{4\ln(4/\beta_m)}{V\widetilde{n}_k^m}} \leq r_k^m - \mu_k \leq \frac{D_m}{VN_m} + \sqrt{\frac{4\ln(4/\beta_m)}{V\widetilde{n}_k^m}}.$$

Additionally, since

$$r_*^m \leq \max_k \left\{ \mu_k + \frac{D_m}{VN_m} + \sqrt{\frac{4\ln(4/\beta_m)}{\widetilde{Vn}_k^m}} - \sqrt{\frac{4\ln(4/\beta_m)}{V\widetilde{n}_k^m}} \right\} \leq \mu_{k^*} + \frac{D_m}{VN_m},$$

$$r_*^m = \max_k \left\{ r_k^m - \sqrt{\frac{4\ln(4/\beta_m)}{V\widetilde{n}_k^m}} \right\} \geq r_{k^*}^m - \sqrt{\frac{4\ln(4/\beta_m)}{V\widetilde{n}_{k^*}^m}} \geq \mu_{k^*} - 2\sqrt{\frac{4\ln(4/\beta_m)}{V\widetilde{n}_{k^*}^m}} - \frac{D_m}{VN_m},$$

we can get

$$-\frac{D_m}{VN_m} - 2\sqrt{\frac{4\ln(4/\beta_m)}{V\widetilde{n}_{k^*}^m}} \leq r_*^m - \mu_{k^*} \leq \frac{D_m}{VN_m}.$$

According to Algorithm 1 and Lemma 2, we have $\widetilde{n}_k^m \geq n_k^m$ for all arms $k$. Then we have the following inequality for all $k \in [K]$:

$$\sqrt{\frac{4\ln(4/\beta_m)}{V\widetilde{n}_k^m}} \leq \sqrt{\frac{4\ln(4/\beta_m)}{Vn_k^m}} = \frac{\Delta_k^{m-1}}{8}.$$

We now establish the upper bound for $\Delta_k^m$ by induction on $m$.

For the base case $m = 1$, the statement is trivial as $\Delta_k^1 = 1$ for all $k \in [K]$.

Assuming the statement is true for the case of $m - 1$, we have

$$\begin{aligned}
\Delta_k^m = r_*^m - r_k^m = (r_*^m - \mu_{k^*}) + (\mu_{k^*} - \mu_k) + (\mu_k - r_k^m) \\
\leq \frac{D_m}{VN_m} + \Delta_k + \frac{D_m}{VN_m} + \frac{1}{8}\Delta_k^{m-1} \\
\leq \frac{2D_m}{VN_m} + \Delta_k + \frac{1}{8}\left(\frac{8\Delta_k}{7} + 2^{-(m-2)} + 2\rho_{m-1}\right) \\
\leq \frac{8\Delta_k}{7} + 2^{-(m-1)} + 2\rho_m,
\end{aligned}$$

Where the second inequality follows from the induction hypothesis.

Next, we provide the lower bound of $\Delta_k^m$. We can get

$$
\begin{aligned}
\Delta_k^m = r_*^m - r_k^m = (r_*^m - \mu_{k^*}) + (\mu_{k^*} - \mu_k) + (\mu_k - r_k^m) \\
\geq -\frac{D_m}{VN_m} - \frac{1}{4}\Delta_{k^*}^{m-1} + \Delta_k - \frac{D_m}{VN_m} - \frac{1}{8}\Delta_k^{m-1} \\
\geq -\frac{2D_m}{VN_m} + \Delta_k - \frac{3}{8}\left(\frac{8\Delta_k}{7} + 2^{-(m-2)} + 2\rho_{m-1}\right) \\
\geq \frac{4}{7}\Delta_k - \frac{3}{2}2^{-m} - 6\rho_m.
\end{aligned}
$$

where the third inequality comes from the upper bound of $\Delta_k^{m-1}$. $\qquad\square$

We only introduces a small change into the parameter $\lambda_m$, by simple calculation, Lemma 3 still holds.

### B.2.3 Proof for Theorem 2

We first define the regret $R_k^m$ generated by arm $k$ in epoch $m$ for agent $v$ as

$$
R_{v,k}^m \triangleq \Delta_k \widetilde{n}_{v,k}^m = \begin{cases} \Delta_k n_k^m & k \neq k_m \\ \Delta_k \widetilde{n}_k^m & k = k_m \end{cases}.
$$

Then we analyze the regret in following three cases:

**Case 1:** $\quad 0 < \Delta_k \leq 64\rho_{m-1}$.
If $k \neq k_m$, then we have

$$
R_{v,k}^m = \Delta_k n_k^m \leq 64\rho_{m-1}n_k^m.
$$

If $k = k_m$, then we have

$$
R_{v,k}^m = \Delta_k \widetilde{n}_{k_m}^m \leq 64\rho_{m-1}N_m.
$$

**Case 2:** $\quad \Delta_k \leq 8 \cdot 2^{-m}$ and $\rho_{m-1} \leq \frac{\Delta_k}{64}$.
If $k \neq k_m$, then we have

$$
R_{v,k}^m = n_k^m\Delta_k = \lambda_m(\Delta_k^{m-1})^{-2}\Delta_k \leq \lambda_m 2^{2(m-1)}\Delta_k \leq \frac{16\lambda_m}{\Delta_k}.
$$

If $k = k_m$, since $\Delta_{k_m}^{m-1} = 2^{-(m-1)}$, we can get:

$$
R_{v,k}^m = \widetilde{n}_{k_m}^m\Delta_{k_m} \leq N_m\Delta_{k_m} = \lceil K\lambda_m 2^{2(m-1)}\rceil\Delta_{k_m} \leq \frac{16K\lambda_m}{\Delta_{k_m}} + \Delta_{k_m} \leq \frac{16K\lambda_m}{\Delta} + 1.
$$

**Case 3:** $\quad \Delta_k > 8 \cdot 2^{-m}$ and $\rho_{m-1} \leq \frac{\Delta_k}{64}$.
By Lemma 7 we have

$$
\Delta_k^{m-1} \geq \frac{4}{7}\Delta_k - \frac{3}{2}2^{-m} - \frac{6}{64}\Delta_k \geq \Delta_k\left(\frac{4}{7} - \frac{3}{16} - \frac{6}{64}\right) \geq 0.29\Delta_k.
$$

In this case, it is impossible that $k = k_m$ because $\Delta_{k_m}^{m-1} = 2^{-(m-1)} < 0.29 \cdot 8 \cdot 2^{-m} < 0.29\Delta_k$.
So we can obtain

$$
R_{v,k}^m = n_k^m\Delta_k = \lambda_m(\Delta_k^{m-1})^{-2}\Delta_k \leq \frac{\lambda_m}{0.29^2\Delta_k} \leq \frac{16\lambda_m}{\Delta_k}.
$$

We define $\mathcal{A}^m \triangleq \left\{ k \in [K] \mid 0 < \Delta_k \le 64\rho_{m-1} \right\}$ for epoch $m$. By combining all three cases, we can upper bound the regret for agent $v$ as

$$
\begin{aligned}
R(T) &= \sum_{m=1}^{M} \left( \sum_{k \in \mathcal{A}^m} R_k^m + \sum_{k \notin \mathcal{A}^m} R_k^m \right) \\
&\le \sum_{m=1}^{M} \Bigg( 64\rho_{m-1}N_m + \sum_{k \in \mathcal{A}^m, k \ne k_m} 64\rho_{m-1}n_k^m + \sum_{k \notin \mathcal{A}^m, \Delta_k > 0} \frac{16\lambda_m}{\Delta_k} \\
&\quad + \left( \frac{16K\lambda_m}{\Delta} + 1 \right) \mathbb{I}(0 < \Delta_{k_m} \le 8 \cdot 2^{-m}) \Bigg) \\
&\le \sum_{m=1}^{M} \Bigg( 64\rho_{m-1}N_m + \sum_{\Delta_k > 0} 64\rho_{m-1}n_k^m + \sum_{\Delta_k > 0} \frac{16\lambda_m}{\Delta_k} \\
&\quad + \left( \frac{16K\lambda_m}{\Delta} + 1 \right) \mathbb{I}\left( m \le \log_2\left(8/\Delta\right) \right) \Bigg) \\
&\le \sum_{m=1}^{M} \left( 128\rho_{m-1}N_m + \sum_{\Delta_k > 0} \frac{16\lambda_m}{\Delta_k} \right) + \sum_{m=1}^{\log_2(8/\Delta)} \left( \frac{16K\lambda_m}{\Delta} + 1 \right)
\end{aligned}
\tag{8}
$$

where the last inequality uses the fact that $\sum_{\Delta_k > 0} n_k^m \le N_m$. Notice that we can bound the expectation of the offset level as

$$
\mathbb{E}[D_m] = 2(1 - \delta_m)C_m + \delta_m N_m \le 2C_m + 1
$$

and we can bound $\sum_{m=1}^{M} \rho_{m-1}N_m$ as

$$
\begin{aligned}
\sum_{m=1}^{M} \rho_{m-1}N_m &\le \sum_{m=1}^{M} \left( \sum_{s=1}^{m-1} \frac{D_s}{8^{m-1-s}VN_s} \right) N_m \\
&= \frac{4}{V} \sum_{m=1}^{M} \left( \sum_{s=1}^{m-1} \frac{(4^{m-1-s}+1)\lambda_m}{8^{m-1-s}\lambda_s} D_s \right) \\
&= \frac{4}{V} \sum_{m=1}^{M} \left( \sum_{s=1}^{m-1} \left( (7/12)^{m-1-s} + (7/48)^{m-1-s} \right) D_s \right) \\
&= \frac{4}{V} \sum_{s=1}^{M-1} D_s \sum_{m=s+1}^{M} (7/12)^{m-1-s} + (7/48)^{m-1-s} \\
&\le \frac{4}{V} \left( \sum_{m=1}^{M-1} D_m \right) \sum_{j=0}^{\infty} (7/12)^j + (7/48)^j \le \frac{11}{V} \sum_{m=1}^{M-1} D_m.
\end{aligned}
\tag{9}
$$

Combining Eq. (8) and Eq. (9), by Lemma 10, we can get

$$
\begin{aligned}
R_v(T) &\leq \sum_{m=1}^{M-1} 1440 \mathbb{E}[D_m] + \sum_{m=1}^{M} \sum_{\Delta_k > 0} \frac{16\lambda_m}{\Delta_k} + \sum_{m=1}^{\log_2(8/\Delta)} \left( \frac{16K\lambda_m}{\Delta} + 1 \right) \\
&\leq \sum_{m=1}^{M-1} 1440 \mathbb{E}[D_m] + \sum_{\Delta_k > 0} \frac{2^{12}(\log^2(VT) + 3\log(VT)\log(30VK))}{V\Delta_k} \\
&\quad + \frac{2^{12}K(\log^2(8/\Delta) + 3\log(8/\Delta)\log(30VK)))}{V\Delta} + \log(8/\Delta) \\
&\leq \sum_{m=1}^{M-1} 2880 \frac{C_m}{V} + 1440 + \sum_{\Delta_k > 0} \frac{2^{14}\log(VT)\log(30VKT)}{V\Delta_k} \\
&\quad + \frac{2^{14}K\log(8/\Delta)\log(240VK/\Delta)}{V\Delta} + \log(8/\Delta) \\
&= O\left( \frac{C}{V} + \sum_{\Delta_k > 0} \frac{\log(T)\log(VKT)}{V\Delta_k} + \frac{K\log(1/\Delta)\log(VK/\Delta)}{V\Delta} \right).
\end{aligned}
$$

So the cumulative regret for all agents as

$$
R(T) = O\left( C + \sum_{\Delta_k > 0} \frac{\log(VT)\log(VKT)}{\Delta_k} + \frac{K\log(1/\Delta)\log(VK/\Delta)}{\Delta} \right).
$$

And each agent only broadcast messages in the end of each epoch, so the communication cost of MA-BARBAT as follows:

$$
\text{Cost}(T) = \sum_{v \in [V]} M = O(V\log(VT)).
$$

### B.3 Proof of Theorem 3

#### B.3.1 Notations

We define $C_k^m$ as the sum of corruptions to arm $k$ in epoch $m$, and let $C_m \triangleq \max_{k \in [K]} C_k^m$.

#### B.3.2 Lemmas for Proving Theorem 3

**Lemma 14.** *For the BB-BARBAT algorithm time horizon $T$, the number of epochs $M = L$. In the $m$-th epoch, the selected arm $k_m$ must satisfy $\Delta_{k_m}^{m-1} = a^{-(m-1)}$.*

*Proof.* The length of epoch $m$ is given by $N_m = K\lambda_m a^{2(m-1)} \geq a^{2(m-1)}$. From the lower bound of $N_m$ and $a = T^{\frac{1}{2(L+1)}}$, we can complete the first statement. Since $\Delta_k^{m-1} \leftarrow \max\{a^{-(m-1)}, r_*^m - r_k^m\}$, there exists at least one arm that satisfies $\Delta_k^{m-1} = a^{-(m-1)}$ and all arms satisfy $\Delta_k^{m-1} \leq a^{-(m-1)}$. Since $r_{k_m}^{m-1} > r_*^{m-1}$, the equality $\Delta_{k_m}^{m-1} = 2^{-(m-1)}$ must hold. □

**Lemma 15.** *For any epoch $m$, the length $N_m$ satisfies $N_m \geq \sum_{k \in [K]} n_k^m$.*

*Proof.* Since $\Delta_k^m = \max\{a^{-m}, r_*^m - r_k^m\} \geq a^{-m}$, we can get $n_k^m = \lambda_m(\Delta_k^{m-1})^{-2} \leq \lambda_m a^{2(m-1)}$. Therefore, we have $\sum_{k \in [K]} n_k^m \leq K\lambda_m a^{2(m-1)} = N_m$. □

**Lemma 16.** *For epoch $s$ and $m$ with $1 \leq s \leq m$, the following inequality holds:*

$$
\frac{\lambda_m}{\lambda_s} \leq \left( \frac{7}{5} \right)^{m-s}.
$$

*Proof.* We first show that the function $f(x) = \frac{x \ln(a^2) + \ln(x)}{(7/5)^x}$ is strictly decreasing for $x \geq 5$, where $a = \max\{T^{\frac{1}{2(L+1)}}, 2\} \geq 2$. Notice that the derivative function

$$f'(x) = \frac{(\ln(a^2) + 1/x) - (x \ln(a^2) + \ln(x)) \ln\left(\frac{7}{5}\right)}{\left(\frac{7}{5}\right)^x}$$

is monotonically decreasing and $f'(5) < 0$, thus we have $f'(x) < 0$ for $x \geq 5$, which indicates that $f(x)$ is strictly decreasing. Since $K \geq 2$, we can get

$$\begin{aligned}
\frac{\lambda_m}{\lambda_s} &= \frac{\ln(4K^2 \ln(K)(m+4)a^{2(m+4)})}{\ln(4K^2 \ln(K)(s+4)a^{2(s+4)})} \\
&= \frac{\ln(4K^2 \ln(K)) + \ln(m+4) + (m+4)\ln(a^2)}{\ln(4K^2 \ln(K)) + \ln(s+4) + (s+4)\ln(a^2)} \\
&< \frac{\ln(m+4) + (m+4)\ln(a^2)}{\ln(s+4) + (s+4)\ln(a^2)} \\
&= \frac{f(m+4)}{f(s+4)} \left(\frac{7}{5}\right)^{m-s} < \left(\frac{7}{5}\right)^{m-s}
\end{aligned}$$

where we use the monotonicity of $f(x)$ and the fact that $K \geq 2$. $\qquad\square$

**Lemma 17.** *For any epoch $m$, the following inequality holds:*

$$\sum_{s=1}^{m} \lambda_s \leq 2a^8 \ln(a)(m^2 + 3m(2 + \ln(K))).$$

*Proof.* Since $a \geq 2$ and $K \geq 2$, we can have

$$\begin{aligned}
\sum_{s=1}^{m} \lambda_s &= \sum_{s=1}^{m} a^8 (\ln(4K^2 \ln(aK)(s+4)a^{2(s+4)})) \\
&= \sum_{s=1}^{m} a^8 (2(s+4)\ln(a) + \ln(s+4) + \ln(4K^2 \ln(aK))) \\
&\leq \sum_{s=1}^{m} a^8 (2(s+4)\ln(a) + 2(s+1)\ln(a) + 6\ln(K)\ln(a)) \\
&\leq \sum_{s=1}^{m} a^8 \ln(a)(4s + 10 + 6\ln(K)) \\
&= 2a^8 \ln(a)(m^2 + 3m(2 + \ln(K))).
\end{aligned}$$

$\qquad\square$

We can also guarantee Lemma 5 still holds. As mentioned before, given the definition of an event $\mathcal{E}_m$ as follows:

$$\mathcal{E}_m \triangleq \left\{ \forall k : |r_k^m - \mu_k| \leq \sqrt{\frac{4\ln(4/\beta_m)}{\widetilde{n}_k^m}} + \frac{2C_m}{N_m} \right\}.$$

Then we can establish a lower bound on the probability of the event $\mathcal{E}_m$ occurring by the following lemma.

**Lemma 18.** *For any epoch $m$, event $\mathcal{E}_m$ holds with probability at least $1 - \delta_m$. We also have $1/\delta_m \geq N_m$.*

*Proof.* By Lemma 5, we can get

$$\Pr\left[ |r_k^m - \mu_k| \leq \sqrt{\frac{4\ln(4/\beta_m)}{\widetilde{n}_k^m}} + \frac{2C_m}{N_m} \right] \leq 2\beta_m = \frac{\delta_m}{K}.$$

A union bound over the $K$ arms conclude the proof.

Since $m \geq 1$, $K \geq 2$ and $a \geq 2$, then we can get

$$
\begin{aligned}
N_m &= K\lambda_m a^{2(m-1)} \\
&= Ka^{2(m+3)}\ln(4K^2(m+4)a^{2(m+4)}\ln(aK)) \\
&\leq Ka^{2(m+3)}(2(m+4)\ln(a) + \ln(m+4) + \ln(K^2\ln(aK)) + \ln(4)) \\
&\leq Ka^{2(m+3)}(2(m+4)\ln(aK) + (m+4)\ln(aK) + 3\ln(aK) + 2\ln(aK)) \\
&\leq Ka^{2(m+3)}((4m+16)\ln(aK)) \\
&= Ka^{2(m+4)}((m+4)\ln(aK)) \\
&= 1/\delta_m.
\end{aligned}
$$

$\square$

As mentioned before, for each epoch $m$, to unify the varying bounds depending on the occurrence of event $\mathcal{E}_m$, we define the offset level

$$
D_m = \begin{cases} 2C_m & \text{when } \mathcal{E}_m \text{ occurs} \\ N_m & \text{when } \mathcal{E}_m \text{ dose not occur} \end{cases}.
$$

This way we can always guarantee the following inequality:

$$
|r_k^m - \mu_k| \leq \sqrt{\frac{4\ln(4/\beta_m)}{\widetilde{n}_k^m}} + \frac{D_m}{N_m}.
$$

It is worth noting that $\Pr[D_m = 2C_m] \geq 1 - \delta_m$ and $\Pr[D_m = N_m] \leq \delta_m$.

Next, we will bound $\Delta_k^m$. To start, we define the discounted offset rate as

$$
\rho_m := \sum_{s=1}^m \frac{2^{m-s}D_s}{a^{4(m-s)}N_s}.
$$

**Lemma 19.** *For all epochs $m$ and arms $k$, we can have*

$$
\frac{a^4-8}{a^4-2}\Delta_k - \frac{3}{2a^2}a^{-m} - 6\rho_m \leq \Delta_k^m \leq \frac{a^4-2}{a^4}\Delta_k + a^{-(m-1)} + 2\rho_m.
$$

*Proof.* Since $|r_k^m - \mu_k| \leq \sqrt{\frac{4\ln(4/\beta_m)}{\widetilde{n}_k^m}} + \frac{D_m}{N_m}$, we have

$$
-\frac{D_m}{N_m} - \sqrt{\frac{4\ln(4/\beta_m)}{\widetilde{n}_k^m}} \leq r_k^m - \mu_k \leq \frac{D_m}{N_m} + \sqrt{\frac{4\ln(4/\beta_m)}{\widetilde{n}_k^m}}.
$$

Additionally, since

$$
r_*^m \leq \max_k\left\{\mu_k + \frac{D_m}{N_m} + \sqrt{\frac{4\ln(4/\beta_m)}{\widetilde{n}_k^m}} - \sqrt{\frac{4\ln(4/\beta_m)}{\widetilde{n}_k^m}}\right\} \leq \mu_{k^*} + \frac{D_m}{N_m},
$$

$$
r_*^m = \max_k\left\{r_k^m - \sqrt{\frac{4\ln(4/\beta_m)}{\widetilde{n}_k^m}}\right\} \geq r_{k^*}^m - \sqrt{\frac{4\ln(4/\beta_m)}{\widetilde{n}_{k^*}^m}} \geq \mu_{k^*} - 2\sqrt{\frac{4\ln(4/\beta_m)}{\widetilde{n}_{k^*}^m}} - \frac{D_m}{N_m},
$$

we can get

$$
-\frac{D_m}{N_m} - \sqrt{\frac{4\ln(4/\beta_m)}{\widetilde{n}_{k^*}^m}} \leq r_*^m - \mu_{k^*} \leq \frac{D_m}{N_m}.
$$

According to Algorithm 1 and Lemma 2, we have $\widetilde{n}_k^m \geq n_k^m$ for all arms $k$. Then we have the following inequality for all $k \in [K]$:

$$
\sqrt{\frac{4\ln(4/\beta_m)}{\widetilde{n}_k^m}} \leq \sqrt{\frac{4\ln(4/\beta_m)}{n_k^m}} = \frac{2\Delta_k^{m-1}}{a^4}.
$$

We now establish the upper bound for $\Delta_k^m$ by induction on $m$.

For the base case $m = 1$, the statement is trivial as $\Delta_k^1 = 1$ for all $k \in [K]$.

Assuming the statement is true for the case of $m - 1$, we have

$$
\begin{aligned}
\Delta_k^m = r_*^m - r_k^m = (r_*^m - \mu_{k^*}) + (\mu_{k^*} - \mu_k) + (\mu_k - r_k^m) \\
\leq \frac{D_m}{N_m} + \Delta_k + \frac{D_m}{N_m} + \frac{2}{a^4}\Delta_k^{m-1} \\
\leq \frac{2D_m}{N_m} + \Delta_k + \frac{2}{a^4}\left(\frac{a^4\Delta_k}{a^4 - 2} + a^{-(m-2)} + 2\rho_{m-1}\right) \\
\leq \frac{a^4\Delta_k}{a^4 - 2} + a^{-(m-1)} + 2\rho_m,
\end{aligned}
$$

Where the second inequality follows from the induction hypothesis.

Next, we provide the lower bound of $\Delta_k^m$. We can get

$$
\begin{aligned}
\Delta_k^m = r_*^m - r_k^m = (r_*^m - \mu_{k^*}) + (\mu_{k^*} - \mu_k) + (\mu_k - r_k^m) \\
\geq -\frac{D_m}{N_m} - \frac{4}{a^4}\Delta_{k^*}^{m-1} + \Delta_k - \frac{D_m}{N_m} - \frac{2}{a^4}\Delta_k^{m-1} \\
\geq -\frac{2D_m}{N_m} + \Delta_k - \frac{6}{a^4}\left(\frac{a^4\Delta_k}{a^4 - 2} + a^{-(m-2)} + 2\rho_{m-1}\right) \\
\geq \frac{a^4 - 8}{a^4 - 2}\Delta_k - \frac{3}{2a^2}a^{-m} - 6\rho_m.
\end{aligned}
$$

where the third inequality comes from the upper bound of $\Delta_k^{m-1}$. $\qquad\square$

### B.3.3 Proof for Theorem 3

We first define the regret $R_k^m$ generated by arm $k$ in epoch $m$ as

$$
R_k^m \triangleq \Delta_k \widetilde{n}_k^m = \begin{cases} \Delta_k n_k^m & k \neq k_m \\ \Delta_k \widetilde{n}_k^m & k = k_m \end{cases}.
$$

Then we analyze the regret in following three cases:

**Case 1:** $0 < \Delta_k \leq 64\rho_{m-1}$.
If $k \neq k_m$, then we have
$$
R_k^m = \Delta_k n_k^m \leq 64\rho_{m-1} n_k^m.
$$

If $k = k_m$, then we have
$$
R_k^m = \Delta_k \widetilde{n}_{k_m}^m \leq 64\rho_{m-1} N_m.
$$

**Case 2:** $\Delta_k \leq 4 \cdot a^{-(m-1)}$ and $\rho_{m-1} \leq \frac{\Delta_k}{64}$.
If $k \neq k_m$, then we have

$$
R_k^m = n_k^m \Delta_k = \lambda_m (\Delta_k^{m-1})^{-2}\Delta_k \leq \lambda_m a^{2(m-1)}\Delta_k \leq \frac{\lambda_m}{\Delta_k}.
$$

If $k = k_m$, since $\Delta_{k_m}^{m-1} = 2^{-(m-1)}$, we can get:

$$
R_k^m = \widetilde{n}_{k_m}^m \Delta_{k_m} \leq N_m \Delta_{k_m} = \lceil K\lambda_m a^{2(m-1)}\rceil\Delta_{k_m} \leq \frac{16K\lambda_m}{\Delta_{k_m}} + \Delta_{k_m} \leq \frac{16K\lambda_m}{\Delta} + 1.
$$

**Case 3:** $\Delta_k > 4 \cdot a^{-(m-1)}$ and $\rho_{m-1} \leq \frac{\Delta_k}{64}$.
By Lemma 7 we have

$$
\Delta_k^{m-1} \geq \frac{a^4 - 8}{a^4 - 2}\Delta_k - \frac{3}{2a^2}a^{-m} - \frac{6}{64}\Delta_k \geq \Delta_k\left(1 - \frac{6}{a^4 - 2} - \frac{3}{8a} - \frac{6}{64}\right) \geq 0.29\Delta_k.
$$

In this case, it is impossible that $k = k_m$ because $\Delta_{k_m}^{m-1} = a^{-(m-1)} < 0.29 \cdot 4 \cdot a^{-(m-1)} < 0.29\Delta_k$. So we can obtain

$$R_k^m = n_k^m \Delta_k = \lambda_m (\Delta_k^{m-1})^{-2}\Delta_k \leq \frac{\lambda_m}{0.29^2 \Delta_k} \leq \frac{16\lambda_m}{\Delta_k}.$$

We define $\mathcal{A}^m \triangleq \{k \in [K] \,\big|\, 0 < \Delta_k \leq 64\rho_{m-1}\}$ for epoch $m$. By combining all three cases, we can upper bound the regret as

$$
\begin{aligned}
R(T) &= \sum_{m=1}^{M} \left( \sum_{k \in \mathcal{A}^m} R_k^m + \sum_{k \notin \mathcal{A}^m} R_k^m \right) \\
&\leq \sum_{m=1}^{M} \left( 64\rho_{m-1}N_m + \sum_{k \in \mathcal{A}^m, k \neq k_m} 64\rho_{m-1}n_k^m + \sum_{k \notin \mathcal{A}^m, \Delta_k > 0} \frac{16\lambda_m}{\Delta_k} \right. \\
&\quad \left. + \left( \frac{16K\lambda_m}{\Delta} + 1 \right) \mathbb{I}(0 < \Delta_{k_m} \leq 4 \cdot a^{-(m-1)}) \right) \\
&\leq \sum_{m=1}^{M} \left( 64\rho_{m-1}N_m + \sum_{\Delta_k > 0} 64\rho_{m-1}n_k^m + \sum_{\Delta_k > 0} \frac{16\lambda_m}{\Delta_k} \right. \\
&\quad \left. + \left( \frac{16K\lambda_m}{\Delta} + 1 \right) \mathbb{I}\left( m \leq \log_a(4/\Delta) + 1 \right) \right) \\
&\leq \sum_{m=1}^{M} \left( 64\rho_{m-1}N_m + \sum_{\Delta_k > 0} 64\rho_{m-1}n_k^m + \sum_{\Delta_k > 0} \frac{16\lambda_m}{\Delta_k} \right. \\
&\quad \left. + \left( \frac{16K\lambda_m}{\Delta} + 1 \right) \mathbb{I}\left( m \leq \log_2(8/\Delta) \right) \right) \\
&\leq \sum_{m=1}^{M} \left( 128\rho_{m-1}N_m + \sum_{\Delta_k > 0} \frac{16\lambda_m}{\Delta_k} \right) + \sum_{m=1}^{\log_2(8/\Delta)} \left( \frac{16K\lambda_m}{\Delta} + 1 \right)
\end{aligned}
\tag{10}
$$

where the last inequality uses the fact that $\sum_{\Delta_k > 0} n_k^m \leq N_m$. Notice that we can bound the expectation of the offset level as

$$\mathbb{E}[D_m] = 2(1 - \delta_m)C_m + \delta_m N_m \leq 2C_m + 1$$

and we can bound $\sum_{m=1}^{M} \rho_{m-1}N_m$ as

$$
\begin{aligned}
\sum_{m=1}^{M} \rho_{m-1}N_m &\leq \sum_{m=1}^{M} \left( \sum_{s=1}^{m-1} \frac{2^{m-1-s}D_s}{a^{4(m-1-s)}N_s} \right) N_m \\
&= a^2 \sum_{m=1}^{M} \left( \sum_{s=1}^{m-1} \frac{((2a^2)^{m-1-s} + 1)\lambda_m}{a^{4(m-1-s)}\lambda_s} D_s \right) \\
&= a^2 \sum_{m=1}^{M} \left( \sum_{s=1}^{m-1} \left( \left( \frac{14}{5a^2} \right)^{m-1-s} + \left( \frac{1}{a^4} \right)^{m-1-s} \right) D_s \right) \\
&= a^2 \sum_{s=1}^{M-1} D_s \sum_{m=s+1}^{M} \left( \frac{14}{5a^2} \right)^{m-1-s} + \left( \frac{1}{a^4} \right)^{m-1-s} \\
&\leq a^2 \left( \sum_{m=1}^{M-1} D_m \right) \sum_{j=0}^{\infty} \left( \frac{14}{5a^2} \right)^j + \left( \frac{1}{a^4} \right)^j \\
&\leq a^2 \left( \frac{5a^2}{5a^2 - 14} + \frac{a^4}{a^4 - 1} \right) \sum_{m=1}^{M-1} D_m \leq \frac{22a^2}{5} \sum_{m=1}^{M-1} D_m.
\end{aligned}
\tag{11}
$$

Combining Eq. (10) and Eq. (11), by Lemma 17, since $a = T^{\frac{1}{2(L+3)}}$ and $M = L$, we can get

$$R(T) \leq \sum_{m=1}^{M-1} 570a^2 \mathbb{E}[D_m] + \sum_{m=1}^{M} \sum_{\Delta_k > 0} \frac{16\lambda_m}{\Delta_k} + \sum_{m=1}^{\log_2(8/\Delta)} \left( \frac{16K\lambda_m}{\Delta} + 1 \right)$$

$$= \sum_{m=1}^{L-1} 570a^2 \mathbb{E}[D_m] + \sum_{m=1}^{L} \sum_{\Delta_k > 0} \frac{16\lambda_m}{\Delta_k} + \sum_{m=1}^{\log_2(8/\Delta)} \left( \frac{16K\lambda_m}{\Delta} + 1 \right)$$

$$\leq \sum_{m=1}^{L-1} 1140a^2 C_m + 570a^2 + 2^5 a^8 \ln(a) \left( \sum_{\Delta_k > 0} \frac{L^2 + 3L(2 + \log(K))}{\Delta_k} \right.$$

$$\left. + \frac{K(\log^2(8/\Delta) + 3\log(8/\Delta)(2 + \log(K)))}{\Delta} \right) + \log_2(8/\Delta)$$

$$= O \left( CT^{\frac{1}{L+3}} + T^{\frac{4}{L+3}} \left( \sum_{\Delta_k > 0} \frac{L\log(KT)}{\Delta_k} + \frac{K\log(T)\log(1/\Delta)\log(K/\Delta)}{L\Delta} \right) \right).$$

### B.3.4 Proof for Theorem 4

*Proof.* In the batched bandit problem, let the static grids be defined as $0 = t_0 < t_1 < \cdots < t_L = T$. According to Lemma 2 in [19], the lower bound for stochastic batched bandits with no corruption are given by:

$$R(C = 0, L) = O \left( K \cdot \max_{j \in [L]} \frac{t_j}{t_{j-1} + 1} \right). \tag{12}$$

For adversarial corruptions, we define an adversary who starts attacking at round 0 and continues until the end of a batch, denoted by $t_s$ for $0 < s < L$. The attack strategy involves setting the rewards of all arms to zero, forcing any algorithm to pull all arms evenly in round $s + 1$, which leads to a regret of approximately $t_s - t_{s-1}$. Therefore, under adversarial corruptions, the lower bound can be expressed as:

$$R(C, L) = O \left( K \cdot \max_{j \in [L]} \frac{t_j}{t_{j-1} + 1} + C \cdot \max_{j \in [L-1]} \frac{t_{j+1} - t_j}{t_j} \right)$$

$$= O \left( Kt_1 + (K + C) \max_{j \in [L-1]} \frac{t_{j+1} - t_j}{t_j} \right) \tag{13}$$

$$= O \left( Kt_1 + (K + C) \left( \frac{T}{t_j} \right)^{\frac{1}{L-1}} \right).$$

According the formula, we can get the optimal value of $t_1 = T^{\frac{1}{L}} \left( \frac{K(L-1)}{K+C} \right)^{\frac{L-1}{L}}$. Combine equation 13, we can get

$$R(C, L) \geq \Omega \left( T^{\frac{1}{L}} \left( K + C^{1 - \frac{1}{L}} \right) \right).$$

$\square$

### B.4 Proof of Theorem 5

#### B.4.1 Lemmas for Proving Theorem 5

By employing the methods described above, we establish that Lemmas 1, 2, 3, and 4 remain valid. In contrast to the standard multi-armed bandit setting, the feedback structure in our model implies that, for each arm $k$, the expected number of observable pulls $\hat{n}_k^m$ does not necessarily equal the expected number of actual pulls $\tilde{n}_k^m$.

**Lemma 20.** *For any epoch $m$, all arms $k \neq k_m$ must satisfy $\tilde{n}_k^m \leq n_k^m$ and $\hat{n}_k^m \geq n_k^m$.*

*Proof.* By Algorithm 4, we can easily guarantee that the inequality $\tilde{n}_k^m \leq n_k^m$ holds for all arms $k \neq k_m$. Recalling the setting of strongly observable graph, for each arm $k$, either it has a self-loop

or all other arms have an edge pointing to it. According to Algorithm 4, we can get the guarantee as follows for each arm $k_j \in [K]$:

$$\sum_{(k_i, k_j) \in E, k_i \neq k_j} \widetilde{n}^m_{k_i} + \widetilde{n}^m_{k_j} \geq n^m_{k_j}.$$

So for each arm $k$ which has a self-loop, we have the following inequality:

$$\hat{n}^m_k = \sum_{(k_i, k_j) \in E, k_i \neq k_j} \widetilde{n}^m_{k_i} + \widetilde{n}^m_{k_j} \geq n^m_k.$$

For each arm $k$ which does not have a self-loop, since $\widetilde{n}^m_{k_m} \geq 2^{-m} \geq n^m_k$ for all arms $k \neq k_m$, so we complete the proof. $\square$

Same as the proof in Appendix B.1, we only need to change $\widetilde{n}^m_k$ to $\hat{n}^m_k$ to obtain the following lemmas.

**Lemma 21.** *For any fixed $k, m$ and $\beta_m$, Algorithm 1 satisfies*

$$\Pr\left[ |r^m_k - \mu_k| \geq \sqrt{\frac{4\ln(4/\beta_m)}{\hat{n}^m_k}} + \frac{2C_m}{N_m} \right] \leq \beta_m.$$

We also define an event $\mathcal{E}_m$ for epoch $m$ as follows:

$$\mathcal{E}_m \triangleq \left\{ \forall\, k : |r^m_k - \mu_k| \leq \sqrt{\frac{4\ln(4/\beta_m)}{\hat{n}^m_k}} + \frac{2C_m}{N_m} \right\}.$$

**Lemma 22.** *For any epoch $m$, event $\mathcal{E}_m$ holds with probability at least $1 - \delta_m$. And after rigorous calculation, we have $1/\delta_m \geq N_m$.*

We can also always guarantee the following inequality:

$$|r^m_k - \mu_k| \leq \sqrt{\frac{4\ln(4/\beta_m)}{\hat{n}^m_k}} + \frac{D_m}{N_m}.$$

It is worth noting that $\Pr[D_m = 2C_m] \geq 1 - \delta_m$ and $\Pr[D_m = N_m] \leq \delta_m$.

Next, we will bound $\Delta^m_k$, which is also the turning point of our proof. To start, we define the discounted offset rate as

$$\rho_m := \sum_{s=1}^{m} \frac{D_s}{8^{m-s} N_s}.$$

**Lemma 23.** *For all epochs $m$ and arms $k \neq k^*$, we can have*

$$\frac{4}{7}\Delta_k - \frac{3}{4}2^{-m} - 6\rho_m \leq \Delta^m_k \leq \frac{8}{7}\Delta_k + 2^{-(m-1)} + 2\rho_m,$$

*and for the optimal arm $k^*$, we have*

$$-\frac{3}{7}\Delta - \frac{3}{4}2^{-m} - 6\rho_m \leq \Delta^m_{k^*} \leq \frac{1}{7}\Delta + 2^{-(m-1)} + 2\rho_m.$$

*Proof.* Since $|r^m_k - \mu_k| \leq \sqrt{\frac{4\ln(4/\beta_m)}{\hat{n}^m_k}} + \frac{D_m}{N_m}$, we have

$$-\frac{D_m}{N_m} - \sqrt{\frac{4\ln(4/\beta_m)}{\hat{n}^m_k}} \leq r^m_k - \mu_k \leq \frac{D_m}{N_m} + \sqrt{\frac{4\ln(4/\beta_m)}{\hat{n}^m_k}}.$$

By Lemma 20, we can get $\hat{n}^m_k \geq n^m_k$ for all arms $k \neq k_m$ and $\hat{n}^m_{k_m} \geq \sum_{k \neq k_m} \widetilde{n}^m_k$ for arm $k_m$. Then we have the following inequalities:

$$\forall k \neq k_m : \sqrt{\frac{4\ln(4/\beta_m)}{\hat{n}^m_k}} \leq \sqrt{\frac{4\ln(4/\beta_m)}{n^m_k}} = \frac{\Delta^{m-1}_k}{8},$$

and

$$\sqrt{\frac{4\ln(4/\beta_m)}{\hat{n}_{k_m}^m}} \leq \max_{k \neq k_m} \sqrt{\frac{4\ln(4/\beta_m)}{n_{k_m}^m}} = \min_{k \neq k_m} \frac{\Delta_k^{m-1}}{8}.$$

Additionally, given that

$$r_*^m \leq \max_k \left\{ \mu_k + \frac{D_m}{N_m} + \sqrt{\frac{4\ln(4/\beta_m)}{\hat{n}_k^m}} - \sqrt{\frac{4\ln(4/\beta_m)}{\hat{n}_k^m}} \right\} \leq \mu_{k^*} + \frac{D_m}{N_m},$$

and

$$\begin{aligned}
r_*^m &= \max_k \left\{ r_k^m - \sqrt{\frac{4\ln(4/\beta_m)}{\hat{n}_k^m}} \right\} \\
&\geq r_{k^*}^m - \sqrt{\frac{4\ln(4/\beta_m)}{\hat{n}_{k^*}^m}} \\
&\geq \mu_{k^*} - \frac{1}{4}\max\{\Delta_{k^*}^{m-1}, \min_{k \neq k_m} \Delta_k^{m-1}\} - \frac{D_m}{N_m} \\
&\geq \mu_{k^*} - \frac{1}{4}\max\{\Delta_{k^*}^{m-1}, \Delta^{m-1}\} - \frac{D_m}{N_m}
\end{aligned}$$

It follows that

$$-\frac{D_m}{N_m} - \frac{1}{4}\max\{\Delta_{k^*}^{m-1}, \Delta^{m-1}\} \leq r_*^m - \mu_{k^*} \leq \frac{D_m}{N_m}.$$

We now establish the upper bound for $\Delta_k^m$ using induction on epoch $m$.

For the base case $m = 1$, the statement is trivial as $\Delta_k^1 = 1$ for all $k \in [K]$.

Assuming the statement is true for $m - 1$, for arm $k \neq k^*$, since there are at least two arms: one optimal arm and one sub-optimal arm, we can obtain

$$\min_{k \neq k_m} \Delta_k^{m-1} \leq \max\{\Delta_{k^*}^{m-1}, \Delta^{m-1}\} \leq \frac{8}{7}\Delta + 2^{-(m-1)} + 2\rho_m.$$

Then we have

$$\begin{aligned}
\Delta_k^m = r_*^m - r_k^m &= (r_*^m - \mu_*) + (\mu_* - \mu_k) + (\mu_k - r_k^m) \\
&\leq \frac{D_m}{N_m} + \Delta_k + \frac{D_m}{N_m} + \frac{1}{8}\max\{\Delta_k^{m-1}, \min_{k \neq k_m} \Delta_k^{m-1}\} \\
&\leq \frac{2D_m}{N_m} + \Delta_k + \frac{1}{8}\left(\frac{8\max\{\Delta_k, \Delta\}}{7} + 2^{-(m-1)} + 2\rho_{m-2}\right) \\
&\leq \frac{8\Delta_k}{7} + 2^{-(m-1)} + 2\rho_m.
\end{aligned}$$

Where the second inequality follows from the induction hypothesis.

For arm $k^*$, we have

$$\begin{aligned}
\Delta_{k^*}^m = r_*^m - r_{k^*}^m &= (r_*^m - \mu_*) + (\mu_* - \mu_{k^*}) + (\mu_{k^*} - r_{k^*}^m) \\
&\leq \frac{D_m}{N_m} + \frac{D_m}{N_m} + \frac{1}{8}\max\{\Delta_{k^*}^{m-1}, \min_{k \neq k_m} \Delta_k^{m-1}\} \\
&\leq \frac{2D_m}{N_m} + \frac{1}{8}\left(\frac{8\Delta}{7} + 2^{-(m-1)} + 2\rho_{m-1}\right) \\
&\leq \frac{1}{7}\Delta + 2^{-(m-1)} + 2\rho_m.
\end{aligned}$$

Next, we establish the lower bound for $\Delta_k^m$. Specifically, for arm $k \neq k^*$ we demonstrate that

$$\Delta_k^m = r_*^m - r_k^m = (r_*^m - \mu_*) + (\mu_* - \mu_k) + (\mu_k - r_k^m)$$

$$\geq -\frac{D_m}{N_m} - \frac{1}{4}\max\{\Delta_{k^*}^{m-1}, \Delta^{k-1}\} + \Delta_k - \frac{D_m}{N_m} - \frac{1}{8}\Delta_k^{m-1}$$

$$\geq -\frac{2D_m}{N_m} + \Delta_k - \frac{3}{8}\left(\frac{8\max\{\Delta_k, \Delta\}}{7} + 2^{-(m-2)} + 2\rho_{m-1}\right)$$

$$\geq \frac{4}{7}\Delta_k - \frac{3}{2}2^{-m} - 6\rho_m.$$

where the third inequality comes from the upper bound of $\Delta_k^{m-1}$.

For arm $k^*$, we have

$$\Delta_{k^*}^m = r_*^m - r_{k^*}^m = (r_*^m - \mu_*) + (\mu_* - \mu_{k^*}) + (\mu_{k^*} - r_{k^*}^m)$$

$$\geq -\frac{D_m}{N_m} - \frac{1}{4}\max\{\Delta_{k^*}^{m-1}, \Delta^{k-1}\} - \frac{D_m}{N_m} - \frac{1}{8}\Delta_{k^*}^{m-1}$$

$$\geq -\frac{2D_m}{N_m} - \frac{3}{8}\left(\frac{8\Delta}{7} + 2^{-(m-2)} + 2\rho_{m-1}\right)$$

$$\geq -\frac{3}{7}\Delta - \frac{3}{2}2^{-m} - 6\rho_m.$$

This proof is complete. $\qquad\square$

**Lemma 24.** *For any strongly observable directed graph $G$ with the independence number $\alpha$, the obtained out-domination set $D$ must satisfy $|D| \leq \lceil\alpha(1 + 2\ln(K/\alpha))\rceil$ by Algorithm 6. Especially, when $G$ is an acyclic graph and undirected graphs, we have $|D| \leq \alpha$.*

*Proof.* If the graph $G$ is undirected, then in each iteration of the Algorithm 6 we select a vertex and remove its neighbors. The selected vertices are therefore pairwise non-adjacent and form an independent set; hence the resulting out-dominating set $D$ satisfies $|D| \leq \alpha$. The remainder of our analysis focuses on the directed graphs.

Recalling the definition of the no-root vertex, for any strongly observable directed graph $G$ with the independence number $\alpha$, we can get that if we remove a no-root vertex $k$ and its out-degree neighbors, the remaining graph $G'$ whose independence number is at most $\alpha - 1$. That's because no vertex in graph $G'$ is connected to the vertex $k$. By the definition of strongly observable directed acyclic graph, we can always get the no-root vertex for the remaining graph, which means that $|D| \leq \alpha$.

Recalling the pseudo-code in Algorithm 6, we denote the loop index as $s$ and initialize $s = 0$. In other words, $G_s = ([V_s], E_s)$ shows the residual graph obtained after $s$ loops of removing vertices. Given any graph $G_s = ([V_s], E_s)$ with no no-root vertices, $G_{s+1}$ is obtained by removing from $G_s$ the vertex $i_s$ with the largest out-degree $d_s^+$. Hence,

$$d_s^+ \geq \frac{|E_S|}{|V_S|} \geq \frac{|V_s|}{2\alpha_s} - \frac{1}{2} \geq \frac{|V_s|}{2\alpha} - \frac{1}{2},$$

by Turan's theorem (e.g., [35]), where $\alpha_s$ is the independence number of $G_s$ and $\alpha \geq \alpha_s$. This shows that

$$|V_{s+1}| = |V_s| - d_s^+ \leq |V_s|\left(1 - \frac{1}{2\alpha}\right) \leq |V_s|e^{-s/2\alpha}.$$

Iterating, we obtain $|V_s| \leq Ke^{-1/2\alpha}$. We define the graph $G_{s_0} = ([V_{s_0}], E_{s_0})$ with the independence number $\alpha_{s_0}$ as the the final residual graph has cycles. We can discuss this in two cases:

**Case 1:** $|V_{s_0}| < \alpha$.

We can get a graph $G_{s_1} = (|V_{s_1}|, E_{s_1})$ with $s_1 \leq s_0$, which satisfies $|V_{s_1}| < \alpha$ and $|V_{s_1-1}| \geq \alpha$. Since $V_{s_1-1} \leq Ke^{-(s_1-1)/(2\alpha)}$, we have $s_1 - 1 \leq \lceil 2\alpha\ln(K/|V_{s_1-1}|)\rceil \leq \lceil 2\alpha\ln(K/\alpha)\rceil$. So we can get

$$|D| \leq |V_{s_1}| + \lceil 2\alpha\ln(K/\alpha)\rceil + 1 \leq \lceil\alpha(1 + 2\ln(K/\alpha))\rceil.$$

**Case 2:** $|V_{s_0}| \geq \alpha$.

By $V_{s_0} \leq K e^{-s_0/(2\alpha)}$, we have $s_0 \leq \lceil 2\alpha \ln(K/|V_{s_0}|) \rceil \leq \lceil 2\alpha \ln(K/\alpha) \rceil$. Because $G_{s_0}$ has no cycles and the independence number $\alpha_{s_0} \leq \alpha$, we can get

$$|D| \leq \alpha_{s_0} + \lceil 2\alpha \ln(K/\alpha) \rceil \leq \lceil \alpha(1 + 2\ln(K/\alpha)) \rceil.$$

The proof is complete. $\qquad\qquad\qquad\qquad\qquad\qquad\qquad\qquad\qquad\qquad\qquad\qquad\qquad\square$

### B.4.2 Proof for Theorem 5

Recalling Lemma 23, for all arms $k \neq k^*$, we have

$$\frac{4}{7}\Delta_k - \frac{3}{4}2^{-m} - 6\rho_m \leq \Delta_k^m \leq \frac{8}{7}\Delta_k + 2^{-(m-1)} + 2\rho_m.$$

We first define the regret $R_k^m$ generated by arm $k$ in epoch $m$ as

$$R_k^m \triangleq \Delta_k \widetilde{n}_k^m = \begin{cases} \Delta_k n_k^m & k \neq k_m \\ \Delta_k \widetilde{n}_k^m & k = k_m \end{cases}.$$

Then we analyze the regret in following three cases:

**Case 1:** $0 < \Delta_k \leq 64\rho_{m-1}$.
If $k \neq k_m$, then we have

$$R_k^m = \Delta_k n_k^m \leq 64\rho_{m-1} n_k^m.$$

If $k = k_m$, then we have

$$R_k^m = \Delta_k \widetilde{n}_{k_m}^m \leq 64\rho_{m-1} N_m.$$

**Case 2:** $\Delta_k \leq 8 \cdot 2^{-m}$ and $\rho_{m-1} \leq \frac{\Delta_k}{64}$.
If $k \neq k_m$, then we have

$$R_k^m = n_k^m \Delta_k = \lambda_m (\Delta_k^{m-1})^{-2} \Delta_k \leq \lambda_m 2^{2(m-1)} \Delta_k \leq \frac{16\lambda_m}{\Delta_k}.$$

If $k = k_m$, since $\Delta_{k_m}^{m-1} = 2^{-(m-1)}$, we can get:

$$R_k^m = \widetilde{n}_{k_m}^m \Delta_{k_m} \leq N_m \Delta_{k_m} = \lceil K\lambda_m 2^{2(m-1)} \rceil \Delta_{k_m} \leq \frac{16K\lambda_m}{\Delta_{k_m}} + \Delta_{k_m} \leq \frac{16K\lambda_m}{\Delta} + 1.$$

**Case 3:** $\Delta_k > 8 \cdot 2^{-m}$ and $\rho_{m-1} \leq \frac{\Delta_k}{64}$.
By Lemma 7 we have

$$\Delta_k^{m-1} \geq \frac{4}{7}\Delta_k - \frac{3}{2}2^{-m} - \frac{6}{64}\Delta_k \geq \Delta_k \left( \frac{4}{7} - \frac{3}{16} - \frac{6}{64} \right) \geq 0.29\Delta_k.$$

In this case, it is impossible that $k = k_m$ because $\Delta_{k_m}^{m-1} = 2^{-(m-1)} < 0.29 \cdot 8 \cdot 2^{-m} < 0.29\Delta_k$. So we can obtain

$$R_k^m = n_k^m \Delta_k = \lambda_m (\Delta_k^{m-1})^{-2} \Delta_k \leq \frac{\lambda_m}{0.29^2 \Delta_k} \leq \frac{16\lambda_m}{\Delta_k}.$$

We define $\mathcal{A}^m \triangleq \{k \in [K] \,\big|\, 0 < \Delta_k \leq 64\rho_{m-1}\}$ for epoch $m$. By combining all three cases, we can upper bound the regret as

$$
\begin{aligned}
R(T) = \sum_{m=1}^{M} & \left( \sum_{k \in \mathcal{A}^m} R_k^m + \sum_{k \notin \mathcal{A}^m} R_k^m \right) \\
\leq \sum_{m=1}^{M} & \left( 64\rho_{m-1}N_m + \sum_{k \in \mathcal{A}^m, k \neq k_m} 64\rho_{m-1}n_k^m + \sum_{k \notin \mathcal{A}^m, \Delta_k > 0} \frac{16\lambda_m}{\Delta_k} \right. \\
& \left. + \left( \frac{16K\lambda_m}{\Delta} + 1 \right) \mathbb{I}(0 < \Delta_{k_m} \leq 8 \cdot 2^{-m}) \right) \\
\leq \sum_{m=1}^{M} & \left( 64\rho_{m-1}N_m + \sum_{\Delta_k > 0} 64\rho_{m-1}n_k^m + \sum_{\Delta_k > 0} \frac{16\lambda_m}{\Delta_k} \right. \\
& \left. + \left( \frac{16K\lambda_m}{\Delta} + 1 \right) \mathbb{I}\left( m \leq \log_2 (8/\Delta) \right) \right) \\
\leq \sum_{m=1}^{M} & \left( 128\rho_{m-1}N_m + \sum_{\Delta_k > 0} \frac{16\lambda_m}{\Delta_k} \right) + \sum_{m=1}^{\log_2(8/\Delta)} \left( \frac{16K\lambda_m}{\Delta} + 1 \right)
\end{aligned}
\tag{14}
$$

where the last inequality uses the fact that $\sum_{\Delta_k > 0} n_k^m \leq N_m$. Notice that we can bound the expectation of the offset level as

$$
\mathbb{E}[D_m] = 2(1 - \delta_m)C_m + \delta_m N_m \leq 2C_m + 1
$$

and we can bound $\sum_{m=1}^{M} \rho_{m-1}N_m$ as

$$
\begin{aligned}
\sum_{m=1}^{M} \rho_{m-1}N_m &\leq \sum_{m=1}^{M} \left( \sum_{s=1}^{m-1} \frac{D_s}{8^{m-1-s}N_s} \right) N_m \\
&\leq 4 \sum_{m=1}^{M} \left( \sum_{s=1}^{m-1} \frac{(4^{m-1-s} + 1)\lambda_m}{8^{m-1-s}\lambda_s} D_s \right) \\
&= 4 \sum_{m=1}^{M} \left( \sum_{s=1}^{m-1} ((7/12)^{m-1-s} + (7/48)^{m-1-s})D_s \right) \\
&= 4 \sum_{s=1}^{M-1} D_s \sum_{m=s+1}^{M} (7/12)^{m-1-s} + (7/48)^{m-1-s} \\
&\leq 4 \left( \sum_{m=1}^{M-1} D_m \right) \sum_{j=0}^{\infty} (7/12)^j + (7/48)^j \leq 11 \sum_{m=1}^{M-1} D_m.
\end{aligned}
\tag{15}
$$

Recalling the pseudo-code of SOG-BARBAT in Algorithm 4, we define the loop (Line 10-16) index as $a$. For ease of analysis, if removing multiple arms from $H^m$ in one loop, we also consider this case as multi loops where the $\overline{H}_a^m$ is set by zero. According to the above analysis, we have $n_k^m \leq \frac{16\lambda_m}{\Delta_k^2}$. Defining $b = K - |\mathcal{A}^m|$, by Lemma 24, we can set a parameter $\gamma^m = \max_{s=1,2,\cdots,b} |D_a^m|$. With loss of generality, for each arm $k \in \mathcal{A}^m$, we set $\Delta_1 \geq \Delta_2 \geq \cdots \geq \Delta_b$. Recalling the process of Algorithm 4, we only need to pull $\overline{H}_a^m$ times for all arms in the out-domination set $D_s^m$, where we can observe at least $\overline{H}_a^m$ times for all arms $k \in [K]$. By this way, to maximize the value of $\sum_{k \in \mathcal{A}^m} R_k^m$, we should try to select the larger part of the suboptimal arm from the arms that have not been removed. In addition, we can have the following inequality:

$$
\overline{H}_1^m \leq n_1^m = \frac{16\lambda_m}{\Delta_1^2}, \overline{H}_2^m - \overline{H}_1^m \leq \frac{16\lambda_m}{\Delta_2^2}, \cdots, \overline{H}_b^m - \overline{H}_{b-1}^m \leq \frac{16\lambda_m}{\Delta_b^2}.
$$

By this way, we can get the following equation:

$$\sup \sum_{k \in \mathcal{A}^m} R_k^m = \sup \sum_{s=1}^{b} \overline{H}_a^m \left( \sum_{k \in \mathcal{A}^m, k \in D_s^m} \Delta_k \right)$$

$$= \sum_{s=1}^{b-\gamma^m+1} \left( \frac{16\lambda_m}{\Delta_s^2} - \frac{16\lambda_m}{\Delta_{s-1}^2} \right) \left( \sum_{k=s}^{s+\gamma^m} \Delta_k \right) + \sum_{s=b-\gamma^m+2}^{b} \left( \frac{16\lambda_m}{\Delta_s^2} - \frac{16\lambda_m}{\Delta_{s-1}^2} \right) \left( \sum_{k=s}^{b} \Delta_k \right)$$

$$= \sum_{k=1}^{\gamma^m} \frac{16\lambda_m}{\Delta_k} + \sum_{k=\gamma^m+1}^{b} \left( \frac{16\lambda_m}{\Delta_k^2} - \frac{16\lambda_m}{\Delta_{k-\gamma^m}^2} \right) \Delta_k$$

$$\leq \sum_{k=\gamma^m+1}^{2\gamma^m} \frac{20\lambda_m}{\Delta_k} + \sum_{k=2\gamma^m+1}^{b} \left( \frac{16\lambda_m}{\Delta_k^2} - \frac{16\lambda_m}{\Delta_{k-1}^2} \right) \Delta_k \leq \cdots \leq \sum_{k=b-\gamma^m+1}^{b} \frac{32\lambda_m}{\Delta_k}.$$

(16)

In the second inequality, for ease of write, we set $\frac{16\lambda_m}{\Delta_0^2} = 0$. For the above inequalities, we use the inequality $\frac{\beta_0}{\Delta_k} + \left( \frac{1}{\Delta_{\gamma^m+k}^2} - \frac{1}{\Delta_k^2} \right) \Delta_{\gamma^m+k} \leq \frac{\beta_1}{\Delta_{\gamma^m+k}}$ holds for all $2 \geq \beta_1 > \beta_0 \geq 1$. For example, in the first inequality, we have

$$\sum_{k=1}^{\gamma^m} \frac{16\lambda_m}{\Delta_k} + \sum_{k=\gamma^m+1}^{2\gamma^m} \left( \frac{16\lambda_m}{\Delta_k^2} - \frac{16\lambda_m}{\Delta_{k-\gamma^m}^2} \right) \Delta_k$$

$$= 16\lambda_m \sum_{k=1}^{\gamma^m} \left( \frac{1}{\Delta_k} + \left( \frac{1}{\Delta_{\gamma^m+k}^2} - \frac{1}{\Delta_k^2} \right) \Delta_{\gamma^m+k} \right) \leq \sum_{k=\gamma^m+1}^{2\gamma^m} \frac{20\lambda_m}{\Delta_k}.$$

Combining Eq. (14), Eq. (15) and Eq. (16), by Lemma 4, we can get

$$R(T) \leq \sum_{m=1}^{M-1} 1440\mathbb{E}[D_m] + \sum_{m=1}^{M} \sum_{\Delta_k>0} \frac{16\lambda_m}{\Delta_k} + \sum_{m=1}^{\log_2(8/\Delta)} \left( \frac{16K\lambda_m}{\Delta} + 1 \right)$$

$$\leq \sum_{m=1}^{M-1} 1440\mathbb{E}[D_m] + \sum_{\Delta_k>0} \frac{2^{12}(\log^2(T) + 3\log(T)\log(30K))}{\Delta_k}$$

$$+ \frac{2^{12}K(\log^2(8/\Delta) + 3\log(8/\Delta)\log(30K)))}{\Delta} + \log(8/\Delta)$$

$$\leq \sum_{m=1}^{M-1} 2880C_m + 1440 + \sum_{\Delta_k>0} \frac{2^{14}\log(T)\log(30KT)}{\Delta_k}$$

$$+ \frac{2^{14}K\log(8/\Delta)\log(240K/\Delta))}{\Delta} + \log(8/\Delta)$$

$$\leq \sum_{m=1}^{M-1} 2880C_m + 1440 + \sum_{k \in \mathcal{I}^*} \frac{2^{15}\log(T)\log(30KT)}{\Delta_k}$$

$$+ \frac{2^{14}K\log(8/\Delta)\log(240K/\Delta))}{\Delta} + \log(8/\Delta)$$

$$= O\left( C + \sum_{k \in \mathcal{I}^*} \frac{\log(T)\log(KT)}{\Delta_k} + \frac{K\log(1/\Delta)\log(K/\Delta)}{\Delta} \right).$$

where $\mathcal{I}^*$ is the set of at most $\lfloor \alpha(1 + 2\ln(K/\alpha)) \rfloor$ arms with the smallest gaps. Especially, for directed acyclic graphs (including undirected graphs), $\mathcal{I}^*$ is the set of at most $\alpha$ arms with the smallest gaps.

## B.5 Proof of Theorem 6

### B.5.1 Notations

Following [15, 14], with loss of generality, we assume $\mu_1 \geq \mu_2 \geq \cdots \geq \mu_K$ and $\Delta_k = \mu_k - \mu_d$ for all arms $k > d$.

### B.5.2 Lemmas for Proving Theorem 6

By the methods mentioned before, Lemma 1 still holds. For Lemma 2, we need to consider the impact of changing $k_m$ representing an arm to representing an arm set.

**Lemma 25.** *For any epoch $m$, the length $N_m$ satisfies $dN_m \geq \sum_{k \in \mathcal{K}} n_k^m$. The actual expected pulling times $\widetilde{n}_k^m \geq n_k^m$ for all arms $k \in [K]$.*

*Proof.* Since $\Delta_k^m = \max\{2^{-m}, r_*^m - r_k^m\} \geq 2^{-m}$, we can get

$$n_k^m = \lambda_m(\Delta_k^{m-1})^{-2} \leq \lambda_m 2^{2(m-1)}.$$

Therefore, we have

$$\sum_{k \in \mathcal{K}} n_k^m \leq K\lambda_m 2^{2(m-1)} = dN_m.$$

Since $N_m = K\lambda_m 2^{2(m-1)}/d$, so for arm $k \in \mathcal{K}_m$, we have

$$\widetilde{n}_k^m = N_m - \sum_{k \notin \mathcal{K}_m} n_k^m/d \geq \lambda_m 2^{2(m-1)} \geq n_k^m.$$

This proof is complete. $\qquad\square$

Using the methods mentioned before, all lemmas in Appendix B.1 still holds. Maybe Lemma 7 will be different because we changed the way $r_*^m$ is assigned. Next we will prove that Lemma 7 still holds.

*Proof.* Since $|r_k^m - \mu_k| \leq \sqrt{\frac{4\ln(4/\beta_m)}{\widetilde{n}_k^m}} + \frac{D_m}{N_m}$, we have

$$-\frac{D_m}{N_m} - \sqrt{\frac{4\ln(4/\beta_m)}{\widetilde{n}_k^m}} \leq r_k^m - \mu_k \leq \frac{D_m}{N_m} + \sqrt{\frac{4\ln(4/\beta_m)}{\widetilde{n}_k^m}}.$$

Additionally, since

$$r_*^m \leq \top_d \left(\left\{\mu_k + \frac{D_m}{N_m} + \sqrt{\frac{4\ln(4/\beta_m)}{\widetilde{n}_k^m}} - \sqrt{\frac{4\ln(4/\beta_m)}{\widetilde{n}_k^m}}\right\}_{k \in [K]}\right) \leq \mu_d + \frac{D_m}{N_m},$$

$$r_*^m = \top_d \left(\left\{r_k^m - \sqrt{\frac{4\ln(4/\beta_m)}{\widetilde{n}_k^m}}\right\}_{k \in [K]}\right) \geq r_d^m - \sqrt{\frac{4\ln(4/\beta_m)}{\widetilde{n}_d^m}} \geq \mu_d - 2\sqrt{\frac{4\ln(4/\beta_m)}{\widetilde{n}_d^m}} - \frac{D_m}{N_m},$$

we can get

$$-\frac{D_m}{N_m} - 2\sqrt{\frac{4\ln(4/\beta_m)}{\widetilde{n}_d^m}} \leq r_*^m - \mu_d \leq \frac{D_m}{N_m}.$$

According to Algorithm 1 and Lemma 2, we have $\widetilde{n}_k^m \geq n_k^m$ for all arms $k$. Then we have the following inequality for all $k \in [K]$:

$$\sqrt{\frac{4\ln(4/\beta_m)}{\widetilde{n}_k^m}} \leq \sqrt{\frac{4\ln(4/\beta_m)}{n_k^m}} = \frac{\Delta_k^{m-1}}{8}.$$

We now establish the upper bound for $\Delta_k^m$ by induction on $m$.

For the base case $m = 1$, the statement is trivial as $\Delta_k^1 = 1$ for all $k \in [K]$.

Assuming the statement is true for the case of $m - 1$, we have

$$\Delta_k^m = r_*^m - r_k^m = (r_*^m - \mu_d) + (\mu_d - \mu_k) + (\mu_k - r_k^m)$$

$$\leq \frac{D_m}{N_m} + \Delta_k + \frac{D_m}{N_m} + \frac{1}{8}\Delta_k^{m-1}$$

$$\leq \frac{2D_m}{N_m} + \Delta_k + \frac{1}{8}\left(\frac{8\Delta_k}{7} + 2^{-(m-2)} + 2\rho_{m-1}\right)$$

$$\leq \frac{8\Delta_k}{7} + 2^{-(m-1)} + 2\rho_m,$$

Where the second inequality follows from the induction hypothesis.

Next, we provide the lower bound of $\Delta_k^m$. We can get

$$\Delta_k^m = r_*^m - r_k^m = (r_*^m - \mu_d) + (\mu_d - \mu_k) + (\mu_k - r_k^m)$$

$$\geq -\frac{D_m}{N_m} - \frac{1}{4}\Delta_{k^*}^{m-1} + \Delta_k - \frac{D_m}{N_m} - \frac{1}{8}\Delta_k^{m-1}$$

$$\geq -\frac{2D_m}{N_m} + \Delta_k - \frac{3}{8}\left(\frac{8\Delta_k}{7} + 2^{-(m-2)} + 2\rho_{m-1}\right)$$

$$\geq \frac{4}{7}\Delta_k - \frac{3}{2}2^{-m} - 6\rho_m.$$

where the third inequality comes from the upper bound of $\Delta_k^{m-1}$. $\qquad\square$

### B.5.3 Proof for Theorem 6

Following to the work [15], we can define the regret $R_k^m$ generated by arm $k$ in epoch $m$ as

$$R_k^m \triangleq \Delta_k \widetilde{n}_k^m = \begin{cases} \Delta_k n_k^m & k \notin \mathcal{K}_m \\ \Delta_k \widetilde{n}_k^m & k \in \mathcal{K}_m \end{cases}.$$

Then we analyze the regret in following three cases:

**Case 1:** $0 < \Delta_k \leq 64\rho_{m-1}$.
If $k \notin \mathcal{K}_m$, then we have

$$R_k^m = \Delta_k n_k^m \leq 64\rho_{m-1}n_k^m.$$

If $k \in \mathcal{K}_m$, then we have

$$R_k^m = \Delta_k \widetilde{n}_{k_m}^m \leq 64\rho_{m-1}dN_m.$$

**Case 2:** $\Delta_k \leq 8 \cdot 2^{-m}$ and $\rho_{m-1} \leq \frac{\Delta_k}{64}$.
If $k \notin \mathcal{K}_m$, then we have

$$R_k^m = n_k^m \Delta_k = \lambda_m(\Delta_k^{m-1})^{-2}\Delta_k \leq \lambda_m 2^{2(m-1)}\Delta_k \leq \frac{16\lambda_m}{\Delta_k}.$$

If $k \in \mathcal{K}_m$, since $\Delta_{k_m}^{m-1} = 2^{-(m-1)}$, we can get:

$$R_k^m = \widetilde{n}_{k_m}^m \Delta_{k_m} \leq dN_m \Delta_{k_m} = d\lceil K\lambda_m 2^{2(m-1)}/d\rceil \Delta_{k_m} \leq \frac{16Kd\lambda_m}{\Delta_{k_m}} + d\Delta_{k_m} \leq \frac{16Kd\lambda_m}{\Delta} + d.$$

**Case 3:** $\Delta_k > 8 \cdot 2^{-m}$ and $\rho_{m-1} \leq \frac{\Delta_k}{64}$.
By Lemma 7 we have

$$\Delta_k^{m-1} \geq \frac{4}{7}\Delta_k - \frac{3}{2}2^{-m} - \frac{6}{64}\Delta_k \geq \Delta_k\left(\frac{4}{7} - \frac{3}{16} - \frac{6}{64}\right) \geq 0.29\Delta_k.$$

In this case, it is impossible that $k \in \mathcal{K}_m$ because $\Delta_{k_m}^{m-1} = 2^{-(m-1)} < 0.29 \cdot 8 \cdot 2^{-m} < 0.29\Delta_k$.
So we can obtain

$$R_k^m = n_k^m \Delta_k = \lambda_m(\Delta_k^{m-1})^{-2}\Delta_k \leq \frac{\lambda_m}{0.29^2\Delta_k} \leq \frac{16\lambda_m}{\Delta_k}.$$

We define $\mathcal{A}^m \triangleq \left\{k \in [K] \,\middle|\, 0 < \Delta_k \le 64\rho_{m-1}\right\}$ for epoch $m$. By combining all three cases, we can upper bound the regret as

$$
\begin{aligned}
R(T) &= \sum_{m=1}^{M} \left( \sum_{k \in \mathcal{A}^m} R_k^m + \sum_{k \notin \mathcal{A}^m} R_k^m \right) \\
&\le \sum_{m=1}^{M} \Bigg( 64\rho_{m-1}dN_m + \sum_{k \in \mathcal{A}^m, k \ne k_m} 64\rho_{m-1}n_k^m + \sum_{k \notin \mathcal{A}^m, \Delta_k > 0} \frac{16\lambda_m}{\Delta_k} \\
&\qquad + \left( \frac{16Kd\lambda_m}{\Delta} + d \right) \mathbb{I}(0 < \Delta_{k_m} \le 8 \cdot 2^{-m}) \Bigg) \\
&\le \sum_{m=1}^{M} \Bigg( 64\rho_{m-1}dN_m + \sum_{\Delta_k > 0} 64\rho_{m-1}n_k^m + \sum_{\Delta_k > 0} \frac{16\lambda_m}{\Delta_k} \\
&\qquad + \left( \frac{16Kd\lambda_m}{\Delta} + d \right) \mathbb{I}\left( m \le \log_2(8/\Delta) \right) \Bigg) \\
&\le \sum_{m=1}^{M} \left( 128\rho_{m-1}dN_m + \sum_{\Delta_k > 0} \frac{16\lambda_m}{\Delta_k} \right) + \sum_{m=1}^{\log_2(8/\Delta)} \left( \frac{16Kd\lambda_m}{\Delta} + d \right)
\end{aligned}
\tag{17}
$$

where the last inequality uses the fact that $\sum_{\Delta_k > 0} n_k^m \le N_m$. Notice that we can bound the expectation of the offset level as

$$
\mathbb{E}[D_m] = 2(1 - \delta_m)C_m + \delta_m N_m \le 2C_m + 1
$$

and we can bound $\sum_{m=1}^{M} \rho_{m-1}N_m$ as

$$
\begin{aligned}
\sum_{m=1}^{M} \rho_{m-1}dN_m &\le \sum_{m=1}^{M} \left( \sum_{s=1}^{m-1} \frac{D_s}{8^{m-1-s}N_s} \right) dN_m \\
&\le 4d \sum_{m=1}^{M} \left( \sum_{s=1}^{m-1} \frac{(4^{m-1-s} + 1)\lambda_m}{8^{m-1-s}\lambda_s} D_s \right) \\
&= 4d \sum_{m=1}^{M} \left( \sum_{s=1}^{m-1} ((7/12)^{m-1-s} + (7/48)^{m-1-s})D_s \right) \\
&= 4d \sum_{s=1}^{M-1} D_s \sum_{m=s+1}^{M} (7/12)^{m-1-s} + (7/48)^{m-1-s} \\
&\le 4d \left( \sum_{m=1}^{M-1} D_m \right) \sum_{j=0}^{\infty} (7/12)^j + (7/48)^j \le 11d \sum_{m=1}^{M-1} D_m.
\end{aligned}
\tag{18}
$$

Combining Eq. (17) and Eq. (18), by Lemma 4, we can get

$$
\begin{aligned}
R(T) &\leq \sum_{m=1}^{M-1} 1440 \mathbb{E}[dD_m] + \sum_{m=1}^{M} \sum_{\Delta_k > 0} \frac{16\lambda_m}{\Delta_k} + \sum_{m=1}^{\log_2(8/\Delta)} \left( \frac{16Kd\lambda_m}{\Delta} + d \right) \\
&\leq \sum_{m=1}^{M-1} 1440 \mathbb{E}[dD_m] + \sum_{\Delta_k > 0} \frac{2^{12}(\log^2(T) + 3\log(T)\log(30K))}{\Delta_k} \\
&\quad + \frac{2^{12}Kd(\log^2(8/\Delta) + 3\log(8/\Delta)\log(30K)))}{\Delta} + d\log(8/\Delta) \\
&\leq \sum_{m=1}^{M-1} 2880 dC_m + 1440d + \sum_{\Delta_k > 0} \frac{2^{14}\log(T)\log(30KT)}{\Delta_k} \\
&\quad + \frac{2^{14}Kd\log(8/\Delta)\log(240K/\Delta)}{\Delta} + d\log(8/\Delta) \\
&= O\left( dC + \sum_{\Delta_k > 0} \frac{\log(T)\log(KT)}{\Delta_k} + \frac{dK\log(1/\Delta)\log(K/\Delta)}{\Delta} \right).
\end{aligned}
$$

|         | BARBAT | DRAA  | IND-BARBAR | IND-FTRL |
|---------|--------|-------|------------|----------|
| K = 12  | 0.13 s | 0.14 s | 0.12 s    | 1.53 s   |
| K = 16  | 0.13 s | 0.15 s | 0.14 s    | 1.91 s   |

Table 2: The cost time of each agent for BARBAT and all baseline methods.

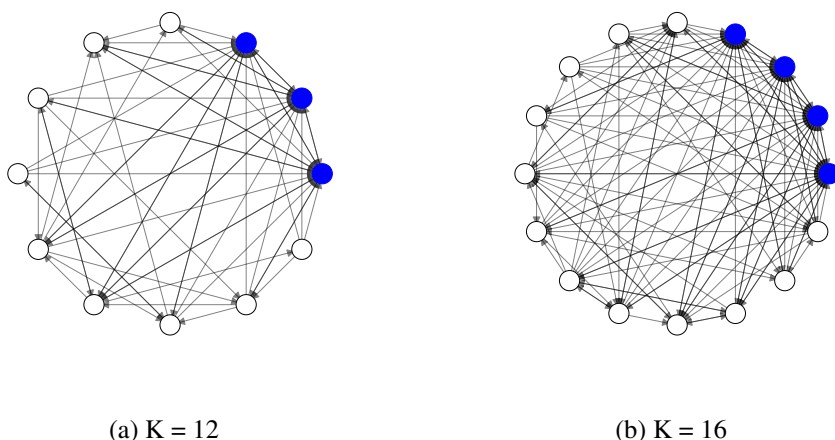

(a) K = 12
(b) K = 16

Figure 4: The feedback structure for the strongly observable graph bandits.

## C   Experimental Details

### C.1   Cooperative Multi-Agent Multi-Armed Bandits

We first consider CMA2B, where the mean rewards $\{\mu_k\}_{k\in[K]}$ is uniformly distributed in the interval $[0.02, 0.96]$. For each arm $k$, we generate the stochastic rewards according to a truncated normal distribution with support $[0, 1]$, mean $\mu_k$, and variance $0.1$. We use a time horizon $T = 50000$, set the corruption level $C = 2000, 5000$ and the number of arms $K = 12, 16$. Following [10], we set the two worst arms as target arms and corrupt the target arm's reward to $1$ and the other arms' rewards to zero until the corruption level is exhausted.

As seen in Table 2, the time cost by BARBAT is much shorter than that of the FTRL, which confirms that the computational efficiency of the BARBAT framework is much higher than that of the FTRL framework.

### C.2   Strongly Observable Graph Bandits

Following [10], we adopt the Erdos–Renyi model [36] to generate feedback graph in Figure 4, with the independence number $\alpha = 4$. Specifically, for each pair of arms $(u, v) \in [K] \times [K]$ with $u = v$, we connect them with a fixed probability $0.5$. Blue circles represent vertices without self-loops, while white circles represent vertices with self-loops.

### C.3   $d$-Set Semi-bandits

In $d$-set semi-bandits, we set $d = 3$ for $K = 12$, and $d = 4$ for $K = 16$. Following the recent works [15, 17], we generate the stochastic rewards according to a Bernoulli distribution, other settings are the same as cooperative standard MAB.

|        | BARBAT | HYBRID | LBINF | LBINF_LS | LBINF_GD |
|--------|--------|--------|-------|----------|----------|
| K = 12 | 0.36 s | 14.63 s | 10.16 s | 10.21 s | 9.84 s |
| K = 16 | 0.37 s | 18.62 s | 12.37 s | 12.67 s | 12.69 s |

Table 3: The cost time of each agent for BARBAT and all baseline methods.

As seen in Table 3, the time cost by BARBAT is much shorter than that of the FTRL, which confirms that the computational efficiency of the BARBAT framework is much higher than that of the FTRL framework.

