# OpenReview forum: "A Near-optimal, Scalable and Parallelizable Framework for Stochastic Bandits Robust to Adversarial Corruptions and Beyond"
_NeurIPS.cc/2025/Conference — NeurIPS 2025 poster_

### Official Review · Reviewer_qGGC · 2025-06-22

**Clarity:** 3
**Significance:** 3
**Originality:** 2
**Rating:** 4
**Confidence:** 3

**Summary:**

The paper focused on the stochastic bandits under the adversarial corruption problem. Based on the limitations of the previous BARBAR algorithm, the authors proposed a new framework called BARBAT based on the elimination idea. Based on the framework, they can get a better regret bound. They also showed the framework on different specific bandit settings, such as multi-agent bandits, graph bandits,
combinatorial semi-bandits and batched bandits, and provided both theoretical and experimental results for these settings.

**Questions:**

1. You provided a lower bound in section 4.2 but how about other settings? Is it possible to provide a lower bound for other settings?
2. From Table 1, the results seem to be a little outperforming previous work. For example, for d-set semmi-bandits, your result is worse than others on the order of $\log T$. Please discuss it.
3. You mentioned the different definition of regret: $R(T)$ and $\tilde{R}(T)$. In the whole paper, you work on $R(T)$, not $\tilde{R}(T)$. Please discuss the motivation for choosing the definition.

**Ethical Concerns:**

["NO or VERY MINOR ethics concerns only"]

**Final Justification:**

I appreciate that the authors provided detailed answers in their response. Especially the lower-bound summary for different settings. My concerns are solved.  Their algorithm is based on the previous method, but they still have solid theoretical results and interesting experimental results. Thus, I would like to keep my score.

**Limitations:**

Yes

**Quality:**

3

**Strengths And Weaknesses:**

Strengths:
The paper improved the previous regret bound for the problem and provided both solid theoretical results and interesting experimental results.

Weaknesses:
The main concern is the novelty of the algorithm since it is based on BARBAR.

---

> ### Author Rebuttal · Authors · 2025-07-30
>
> Thank you for your time and helpful suggestions. We address the reviewer’s concerns as follows.
>
> >The main concern is the novelty of the algorithm since it is based on BARBAR.
>
> Though our method is based on BARBAR, we want to emphsize the novelty of our BARBAT algorithm.
> Unlike BARBAR, which sets the epoch length to $N_m = \sum_{k} n_k^m$, our BARBAT algorithm defines the epoch length as $ N_m \approx O(K 2^{2(m-1)}) $, making it unaffected by the adversary. However, this change introduces challenges in algorithm design and regret analysis since we have $ N_m > \sum_{k} n_k^m $  and require addtional pulling of arms. To address this issue, we increase the probability of selecting the estimated best arm $k_m$. However, pulling the estimated optimal arm $k_m$ too many times may lead to additional regret, as $k_m$ could be suboptimal. To mitigate this potential regret, we adopt epoch-varying failure probabilities $ \delta_m \approx O(1/(mK2^{2m})) $ and demonstrate that such additional regret will only manifest in the first $ O(\log(1/\Delta))$ epochs, thus remaining bounded by a term independent of $T$. Overall, the additional pulls of the estimated best arm in our algorithm, along with the corresponding regret analysis, is  novel and nontrivial. By utilizing these techniques, our algorithms achieve better regret bounds than that of BARBAR.
>
> >You provided a lower bound in section 4.2 but how about other settings? Is it possible to provide a lower bound for other settings?
>
> Thank you for the valuable suggestions. The following table shows the lower bounds of all extensions. The lower bound for multi-armed bandits comes from Gupta et al. (2019), and their results can be directly extended to d-semi bandits, multi-agent bandits, and strongly observable graph bandits by incorporating the lower bounds for stochastic settings (without corruption). However, the lower bound for batch bandits cannot be directly derived from the results of Gupta et al. (2019), so we provide a proof in our paper. We will include all lower bounds in the revision.
>
> | Setting | Lower bounds | Our upper bound |
> |:-------:|:-------:|:-------:|
> | Multi-arm bandits | $C + \sum_{\Delta_k>0}\frac{\log(T)}{\Delta_k}$ | $C + \sum_{\Delta_k>0}\frac{\log(T)^2}{\Delta_k}$ |
> | Multi-agent multi-arm bandits bandits | $\frac{C}{V} + \sum_{\Delta_k>0}\frac{\log(T)}{V\Delta_k}$ | $\frac{C}{V} + \sum_{\Delta_k>0}\frac{\log^2(T)}{V\Delta_k}$ |
> | Batched bandits | $T^{\frac{1}{\mathcal{M}}}(K + C^{1 - \frac{1}{\mathcal{M}}})$ (this paper) | $C T^{\frac{1}{M+3}} + T^{\frac{4}{M+3}} ( \sum_{\Delta_k > 0} \frac{M \log(T)}{\Delta_k})$ |
> | Strongly observable graph bandits | $C + \sum_{k \in \mathcal{I}^*}\frac{\log(T)}{\Delta_{k}}$ | $C + \sum_{k \in \mathcal{I}^*}\frac{\log^2(T)}{\Delta_{k}}$ |
> | $d$-set semi bandits | $dC + \sum_{\Delta_k > 0}\frac{\log(T)}{\Delta_{k}}$ | $dC + \sum_{\Delta_k > 0}\frac{\log(T)}{\Delta_{k}}$ |
>
>
>
> >From Table 1, the results seem to be a little outperforming previous work. For example, for $d$-set semmi-bandits, your result is worse than others on the order of  $\log(T)$. Please discuss it.
>
> * The optimal regret bound for semi-bandits is achieved using the FTRL framework (Tsuchiya et al., 2023). In Section 7 of their paper, Tsuchiya et al. note that one limitation of their algorithm is its computational complexity, as the sampling probabilities cannot be efficiently computed. In contrast, our approach is computationally efficient, although the regret is slightly worse than state-of-the-art results.
>
> * For the multi-agent setting, our regret improvement is also non-incremental. Notice that the regret bound of DRAA is $O(C/V + K\log^2(T)/(V\Delta))$ while the regret bound of BARBAT is $O(C/V + (1/V)\sum_{\Delta_k>0}\log^2(T)/\Delta_k)$. When the minimal gap $\Delta$ is significantly smaller than other suboptimal gaps $\Delta_k\neq \Delta$, our bound may substantially improves upon DRAA. For instance, consider the senario where mean rewards is $1, 1-\Delta,0,\dots,0$. Then DRAA incurs regret $C/V+K\log^2(T)/(V\Delta)$, while our regret is only $C/V+(1/V)(1/\Delta + K)\log^2(T)$. In particular, when $\Delta \leq 1/K$, the corruporion-independent term in the regret bound of DRAA is effectively $K$ times that of BARBAT.
>
> * For the batched bandits, our work is the first to study batched bandits with adversarial corruptions and achieve near-optimal regret bounds.
>
> >You mentioned the different definition of regret: $R(T)$ and $\tilde R(T)$. In the whole paper, you work on $R(T)$, not $\tilde R(T)$. Please discuss the motivation for choosing the definition.
>
> $R(T) = T\mu_{k^*} - \mathbf{E}[\sum_{t=1}^T \mu_{k_t}]$ is a standard regret defnition in stochastic bandits, while the regret definition $\tilde R(T) = \max_{k \in [K]}\mathbf{E}[\sum_{t=1}^T \tilde r_{t,k} - \sum_{t=1}^T \tilde r_{t,I_t}]$ is typically used in adversarial bandits, where $\tilde r_{k}$ represents the corrupted rewards. We think that comparing against the true means, rather than the corrupted rewards, is more natural. Also, $R(T)$ has been employed in all previous works on adversarial corruptions.
>
> References:
> * Anupam Gupta, Tomer Koren, and Kunal Talwar. Better algorithms for stochastic bandits with adversarial corruptions. In Conference on Learning Theory, pages 1562–1578. PMLR, 2019.
> * Taira Tsuchiya, Shinji Ito, and Junya Honda. Further adaptive best-of-both-worlds algorithm for combinatorial semi-bandits. In International Conference on Artificial Intelligence and Statistics, pages 8117–8144. PMLR, 2023.

---

> > ### Comment · Reviewer_qGGC · 2025-08-03
> >
> > Thank you for your answer. It will help me assess the work better.

---

> ### Author Response · Authors · 2025-08-03
>
> Thank you for the reply. We are glad that our response has helped you better understand our paper. We will incorporate your suggestions into the revision.

---

### Official Review · Reviewer_o47x · 2025-06-25

**Clarity:** 4
**Significance:** 3
**Originality:** 2
**Rating:** 4
**Confidence:** 3

**Summary:**

This paper proposes BARBAT, an improved algorithmic framework for stochastic bandits robust to adversarial corruptions. Based on the BARBAR algorithm, BARBAT achieves a near-optimal regret bound of $O(C+ \sum_{\Delta_k>0} \log T^2/\Delta_k)$, removing the suboptimal $O(KC)$ term. The paper also extends BARBAT to settings including multi-agent, graph-structured, semi-bandits, and batched bandits, which is a naturally generalization for successive elimination algorithms.

**Questions:**

See weaknesses

**Ethical Concerns:**

["NO or VERY MINOR ethics concerns only"]

**Final Justification:**

The authors have provided detailed response to my concerns. While the paper is well-presented and the problem is interesting, their theoretical contribution is not very exciting. Therefore, after consideration, I am maintaining my score of 4.

**Limitations:**

See weaknesses

**Paper Formatting Concerns:**

The paper complies with the NeurIPS 2025 formatting guidelines. No issues observed.

**Quality:**

3

**Strengths And Weaknesses:**

Strengths:
This paper is well-written and easy to follow, with clear and precise technical discussions. It extends the core framework to a wide range of bandit settings and is supported by extensive numerical experiments. The breadth of the work reflects a commendable level of effort and thoroughness by the authors.

Weaknesses:
The proposed BARBAT algorithm appears structurally similar to BARBAR, with the main change being in the change of epoch lengths. If achieving the improved regret bounds requires nontrivial technical innovations beyond modifying this aspect, I encourage the authors to highlight and clarify them. One of the primary advantages claimed is BARBAT's scalability and parallelizability, which are indeed important. However, these properties are common to many elimination-based methods, and it is unclear whether they stem from novel design choices in BARBAT. If direct generalization is not sufficient for the multiple settings considered, it would be helpful for the authors to elaborate on the challenges involved and the innovations introduced. Overall, while the work is executed with care, its contributions feel somewhat incremental.

---

> ### Author Rebuttal · Authors · 2025-07-30
>
> Thanks for your detailed review of our work. Here are our response to your comments.
>
> >The proposed BARBAT algorithm appears structurally similar to BARBAR, with the main change being in the change of epoch lengths. If achieving the improved regret bounds requires nontrivial technical innovations beyond modifying this aspect, I encourage the authors to highlight and clarify them.
>
> Unlike BARBAR, which sets the epoch length to $ N_m = \sum_{k} n_k^m $, our BARBAT algorithm defines the epoch length as $ N_m \approx O(K 2^{2(m-1)}) $, making it unaffected by the adversary. However, this change introduces challenges in algorithm design and regret analysis since we have $ N_m > \sum_{k} n_k^m $ and require addtional pulling of arms. To address this issue, we increase the probability of selecting the estimated best arm $k_m$. However, pulling the estimated optimal arm $k_m$ too many times may lead to additional regret, as $k_m$ could be suboptimal. To mitigate this potential regret, we adopt epoch-varying failure probabilities $ \delta_m \approx O(1/(mK2^{2m})) $ and demonstrate that such additional regret will only manifest in the first $ O(\log(1/\Delta))$ epochs, thus remaining bounded by a term independent of $T$. Overall, the additional pulls of the estimated best arm in our algorithm, along with the corresponding regret analysis, represent a novel and nontrivial contribution.
>
> >One of the primary advantages claimed is BARBAT's scalability and parallelizability, which are indeed important. However, these properties are common to many elimination-based methods, and it is unclear whether they stem from novel design choices in BARBAT. If direct generalization is not sufficient for the multiple settings considered, it would be helpful for the authors to elaborate on the challenges involved and the innovations introduced.
>
> Direct generalization of existing approaches is not sufficient for the extensions. Our algorithmic adjustments for each extension are thoughtfully designed to address the specific challenges while maintaining the core advantages of the framework:
> * In strongly observable graph bandits, the main challenge is how to leverage the graph-structured feed-back to reduce the regret suffered from exploration. We introduce the novel OODS algorithm (in Appendix A) for action-probability allocation. Specifically, we iteratively compute a minimum dominating set on the communication graph $G$, assigning pulling probabilities accordingly. Once an arm accumulates sufficient observations, we remove it from the graph and recalculate the dominating set. This process repeats until the graph is empty.
> * In batched bandits, the main challenge here is that the agent can only observe rewards after each batch is completed (instead of after each pull) and the number of batches is limited. To address this, we introduce a batch-related parameter $a = \max\\{T^{\frac{1}{2(\mathcal{M} + 1)}}, 2\\}$, where $\mathcal{M}$ is the number of batches. This parameter $a$ is used to design the exploration parameter $\lambda_m$, the epoch length $N_m$, and the estimated gap $\Delta_k^m$ in our algorithm for batched bandits.
> * In the $d$-set semi-bandit setting, the primary challenge lies in identifying an optimal action composed of $d$ arms, rather than merely a single optimal arm. To address this, we introduce a novel approach: when computing the epoch threshold $r_*^m$, we specifically consider the $d$-th largest empirical reward among arms. This innovation intuitively captures that each arm within the optimal action set can be regarded as an "optimal arm," ensuring effective identification of the optimal $d$-arm combination.
> * For the multi-agent setting, we agree that many elimination-based methods also have parallelizability, but our method achieves state-of-the-art regret bounds.

---

> > ### Comment · Reviewer_o47x · 2025-08-04
> >
> > Thank you for addressing my concerns. While I appreciate the clarifications provided, I still find the theoretical contribution to be somewhat incremental. Therefore, I will maintain my original score.

---

> > > ### Author Response · Authors · 2025-08-06
> > >
> > > Thank you for the reply. We are glad that our response has helped you better understand our paper. We will incorporate your suggestions into the revision.

---

### Official Review · Reviewer_dQYX · 2025-07-01

**Clarity:** 4
**Significance:** 2
**Originality:** 3
**Rating:** 5
**Confidence:** 2

**Summary:**

This paper proposes BARBAT, a novel algorithmic framework for stochastic multi-armed bandits (MABs) under adversarial corruptions. BARBAT improves upon the BARBAR algorithm [1]  by eliminating the suboptimal corruption-dependent term $\mathcal{O}(KC)$  in favor of a near-optimal $\mathcal{O}(C)$ regret bound up to logarithmic factors, where $K$ is the number of arms and $C$ is the corruption budget. The authors further demonstrate the versatility of the framework by extending it to multi-agent, batched, strongly observable graph, and d-set semi-bandit settings, providing tailored algorithms and regret analyses for each. The theoretical results are complemented by empirical validations, showing that BARBAT and its extensions outperform prior methods in several regimes.

[1] Anupam Gupta, Tomer Koren, and Kunal Talwar. Better algorithms for stochastic bandits with adversarial corruptions. In Conference on Learning Theory, pages 1562–1578. PMLR, 2019.

**Questions:**

Please see the Weaknesses section for related questions and concerns.

**Ethical Concerns:**

["NO or VERY MINOR ethics concerns only"]

**Final Justification:**

The authors responded appropriately to my questions and provided the requested lower bounds for the various settings.

**Limitations:**

yes

**Paper Formatting Concerns:**

No major concern

**Quality:**

3

**Strengths And Weaknesses:**

Below is a summary of the paper’s main strengths and weaknesses.

**Strengths**:

1) The paper successfully generalizes BARBAT to four complex and practically relevant settings. The algorithmic adjustments for each extension are thoughtfully designed and maintain the core advantages of the framework. For each setting, the paper provides tight regret upper bounds.

2)  BARBAT avoids the need to solve per-round convex optimization problems (as required by FTRL-based methods), making it computationally efficient.

3) The introduction of BB-BARBAT constitutes the first theoretical analysis of batched bandits under adversarial corruptions, along with a corresponding lower bound. This fills a meaningful gap in the literature.

4)  The experimental results across multiple settings consistently demonstrate that BARBAT-based methods outperform baseline algorithms. The results also show robustness to increasing corruption budgets, further supporting the theoretical claims.

**Weaknesses**:

1) The core BARBAT algorithm is closely related to DRAA [2], with the primary distinction being the use of epoch-varying failure probabilities. While this yields slightly improved regret bounds, this improvement is quite incremental.


2) The multi-agent setting assumes instantaneous, reliable, and cost-free broadcasts. This assumption, while common in theory, limits the practical relevance of the results, especially since the authors claim their method to be highly parallelizable. There is no discussion of asynchronous or faulty communication scenarios, which are common in real-world systems.  Could the framework be extended to handle asynchronous message passing?

3) While a lower bound is provided for batched bandits, no new lower bounds are given for other extensions (e.g., multi-agent or semi-bandits), making it unclear how tight the regret is in these settings.  Is it possible to derive lower bounds for these setting?

[2] Fatemeh Ghaffari, Xuchuang Wang, Jinhang Zuo, and Mohammad Hajiesmaili. Multi-agent stochastic bandits robust to adversarial corruptions. arXiv preprint arXiv:2411.08167, 2024

---

> ### Author Rebuttal · Authors · 2025-07-29
>
> Thank you for taking the time to review our paper. We hope that our response below addresses your concerns.
>
> >The core BARBAT algorithm is closely related to DRAA [2], with the primary distinction being the use of epoch-varying failure probabilities. While this yields slightly improved regret bounds, this improvement is quite incremental.
>
> We do not agree that our regret improvement is quite incremental. Notice that in the MAB setting, the regret bound of DRAA is $O(C + K\log^2(T)/\Delta)$ while the regret bound of BARBAT is $O(C + \sum_{\Delta_k>0}\log^2(T)/\Delta_k)$. When the minimal gap $\Delta$ is significantly smaller than other suboptimal gaps $\Delta_k\neq \Delta$, our bound may substantially improves upon DRAA. For instance, consider the senario where mean rewards is $1, 1-\Delta,0,\dots,0$. Then DRAA incurs regret $C+K\log^2(T)/\Delta$, while our regret is only $C+(1/\Delta + K)\log^2(T)$. In particular, when $\Delta \leq 1/K$, the corruption-independent term in the regret bound of DRAA is effectively $K$ times that of BARBAT.
>
> >The multi-agent setting assumes instantaneous, reliable, and cost-free broadcasts. This assumption, while common in theory, limits the practical relevance of the results, especially since the authors claim their method to be highly parallelizable. There is no discussion of asynchronous or faulty communication scenarios, which are common in real-world systems. Could the framework be extended to handle asynchronous message passing?
>
> We believe it is possible to extend our framework to an asynchronous setting, though it may encounter some challenges. Here is a rough outline of our approach: each agent independently executes the single-agent BARBAT algorithm, while incorporating an early-stopping mechanism. Specifically, an agent will enter the next epoch as soon as it has observed sufficient reward samples—either through its own pulls or those received from other agents—to reliably estimate the reward gaps. A key challenge in this approach is determining the optimal timing for broadcasting the agents' local information. Broadcasting too early may result in insufficient data collection, necessitating additional rounds of communication. Conversely, delaying broadcasts might lead to redundant pulls, preventing the agents from fully leveraging the benefits of collaboration.
>
> >While a lower bound is provided for batched bandits, no new lower bounds are given for other extensions (e.g., multi-agent or semi-bandits), making it unclear how tight the regret is in these settings. Is it possible to derive lower bounds for these setting?
>
> Thank you for the valuable suggestions. The following table shows the lower bounds of all extensions. The lower bound for multi-armed bandits comes from Gupta et al. (2019), and their results can be directly extended to d-semi bandits, multi-agent bandits, and strongly observable graph bandits by incorporating the lower bounds for stochastic settings (without corruption). However, the lower bound for batch bandits cannot be directly derived from the results of Gupta et al. (2019), so we provide a proof in our paper. We will include all lower bounds in the revision.
>
> | Setting | Lower bounds | Our upper bound |
> |:-------:|:-------:|:-------:|
> | Multi-arm bandits | $C + \sum_{\Delta_k>0}\frac{\log(T)}{\Delta_k}$ | $C + \sum_{\Delta_k>0}\frac{\log(T)^2}{\Delta_k}$ |
> | Multi-agent multi-arm bandits bandits | $\frac{C}{V} + \sum_{\Delta_k>0}\frac{\log(T)}{V\Delta_k}$ | $\frac{C}{V} + \sum_{\Delta_k>0}\frac{\log^2(T)}{V\Delta_k}$ |
> | Batched bandits | $T^{\frac{1}{\mathcal{M}}}(K + C^{1 - \frac{1}{\mathcal{M}}})$ (this paper) | $C T^{\frac{1}{M+3}} + T^{\frac{4}{M+3}} ( \sum_{\Delta_k > 0} \frac{M \log(T)}{\Delta_k})$ |
> | Strongly observable graph bandits | $C + \sum_{k \in \mathcal{I}^*}\frac{\log(T)}{\Delta_{k}}$ | $C + \sum_{k \in \mathcal{I}^*}\frac{\log^2(T)}{\Delta_{k}}$ |
> | $d$-set semi bandits | $dC + \sum_{\Delta_k > 0}\frac{\log(T)}{\Delta_{k}}$ | $dC + \sum_{\Delta_k > 0}\frac{\log(T)}{\Delta_{k}}$ |
>
>
> References:
> * Anupam Gupta, Tomer Koren, and Kunal Talwar. Better algorithms for stochastic bandits with adversarial corruptions. In Conference on Learning Theory, pages 1562–1578. PMLR, 2019.

---

> > ### Comment · Reviewer_dQYX · 2025-08-04
> >
> > I thank the author for the informative response. My questions are all well answered.

---

> > > ### Author Response · Authors · 2025-08-06
> > >
> > > Thank you for the reply. We are pleased to have addressed your concerns and will incorporate your suggestions into the revision.

---

### Official Review · Reviewer_FSFH · 2025-07-01

**Clarity:** 3
**Significance:** 3
**Originality:** 3
**Rating:** 4
**Confidence:** 2

**Summary:**

This study investigates the problem of stochastic bandits with adversarial corruptions. The authors refine the BARBAR algorithm and resolve an open issue presented in Gupta et al. Under the refined algorithm, the regret bounds are improved across various settings. The advantage of this algorithm, compared with other algorithms such as Tsallis-INF, lies in its computational efficiency.

**Questions:**

See above.

**Ethical Concerns:**

["NO or VERY MINOR ethics concerns only"]

**Final Justification:**

As I replied to the authors, they have fully addressed my concerns. I believe that this study makes solid contributions. However, I personally find them not particularly surprising. I will maintain my original score, weak accept.

**Limitations:**

N/A.

**Paper Formatting Concerns:**

N/A.

**Quality:**

3

**Strengths And Weaknesses:**

The authors provide a general framework for stochastic bandits with adversarial corruptions. Although several algorithms have been proposed, many of them are computationally costly, which is undesirable for settings such as graph or batched bandits. For example, $(1/2)$-Tsallis-INF requires solving an optimization problem at every step, which makes parallelization difficult.

BARBAR was proposed by Gupta et al. and is known to be computationally efficient. However, its regret does not match the lower bound with respect to the number of arms $K$ and the corruption level $C$. The authors refine the BARBAR algorithm and extend it to the settings mentioned above. They show that the new algorithm achieves favorable performance while retaining computational efficiency.

I believe the authors’ contribution is worth accepting to the conference; however, I still have some concerns. The main issue is that the regret bound is not tight in $T$. For example, if the algorithm incurs $\log(T)^2$ regret in the stochastic bandit setting, one could instead use Shannon-FTRL, which also achieves $\log(T)^2$ regret and is computationally efficient.

Minor comment: In the abstract, the authors use $K$ and $C$ without definition. Although the meaning can be inferred, it may be confusing.

---

> ### Author Rebuttal · Authors · 2025-07-29
>
> Thanks for your detailed review of our work. Here are our response to your comments.
> >The main issue is that the regret bound is not tight in $T$. For example, if the algorithm incurs $\log(T)^2$ regret in the stochastic bandit setting, one could instead use Shannon-FTRL, which also achieves $\log(T)^2$ regret and is computationally efficient.
>
> We acknowledge that $\log(T)^2$ regret is not tight, and we believe that finding a way to achieve optimal regret for adversarial corruptions using elimination-based methods is an interesting open problem. We think our work has made significant strides toward addressing this issue.
>
> Though Shannon-FTRL also achieves $\log(T)^2$ regret and is computationally efficient in the stochastic MAB setting,  our method outperforms Shannon-FTRL in many extended scenarios. In the strong observation graph bandit setting, Shannon-FTRL achieves only $\log(T)^3$ regret, which is worse than the regret bound attained by our method. In the $d$-sets bandit problem, to the best of our knowledge, there is no computationally efficient FTRL-type robust algorithm. As pointed by Tsuchiya et al. (2023), the computational cost is the main limitation of FTRL framework, especially when the optimization problem does not have a closed form. Additionally, for the batched bandit setting, our work provides the first algorithm that is robust to adversarial corruption.
>
> > In the abstract, the authors use $K$ and $C$ without definition. Although the meaning can be inferred, it may be confusing.
>
> Thank you for the comment. We will include the definitions of $K$ and $C$ in the abstract.
>
> References:
> * Shinji Ito, Taira Tsuchiya, and Junya Honda. Nearly optimal best-of-both-worlds algorithms for online learning with feedback graphs. NeurIPS, 2022.
> * Taira Tsuchiya, Shinji Ito, and Junya Honda. Further adaptive best-of-both-worlds algorithm for combinatorial semi-bandits. In ICML, 2023.

---

> > ### Comment · Reviewer_FSFH · 2025-08-04
> > **Reply to the authors' rebuttal**
> >
> > Thank you for your reply. The authors have fully addressed my concerns. I believe that this study makes solid contributions, although I personally find them not particularly surprising. I will maintain my original score.

---

> > > ### Author Response · Authors · 2025-08-06
> > >
> > > Thank you for the reply. We are pleased to have addressed your concerns and will incorporate your suggestions into the revision.

---

### Decision · Program_Chairs · 2025-09-17

**Decision:**

Accept (poster)

**Comment:**

(a) Summary: This work proposed BARBAT, an improved algorithmic framework for stochastic bandits robust to adversarial corruptions. Based on the BARBAR algorithm, BARBAT achieves a near-optimal regret bound of $O(C+\sum_{\Delta_k>0}\log T^2/\Delta_k )$, which removes the suboptimal $O(KC)$ term. The work also extended BARBAT to settings including multi-agent, graph-structured, semi-bandits, and batched bandits, which is a naturally generalization for successive elimination algorithms.

(b) Main reasons for decision/Pros: The work proposed BARBAT, which improved the popular BARBAR algorithm and clarified its contribution.

(c) Change during rebuttal: A comparison among theoretical lower bounds and upper bounds is provided in the reply to reviewer dQYX.

(d) Decision: I vote for accept at the moment but I am not certain about that.
- Two reviewers suggested that the technical challenge is a bit incremental and voted for 4, while the remaining reviewer (reviewer dQYX) voted for 5 but pointed out that he/she is not an expert in the topic of bandits with corruption.